# Stabilizing dual-phased perovskite towards high performance photovoltaics with enhanced batch stability and consistency

Guihua Zhang[1,2,3,8], Deng Wang[1,4,8], Bowei Li [5,8], Qing Lian [1,8] ✉, Xinyi Zou[6], Dongyang Li [1], Qiming Yin [1], Guojun Mi[1], Jie Li[1], Kui Feng [1], Abbas Amini [7], Alex. K. -Y. Jen [4], Xugang Guo [1], Baomin Xu [1] ✉ & Chun Cheng [1,2,3] ✉

Fabricating high-performance perovskite solar cells (PSCs) with solution processing is conducive to low-cost commercial production, it is therefore rather critical to stabilize perovskite in both solution and solid phases. For this purpose, the speed-up ageing of perovskite solution in air was systematically studied and its severe spontaneous degradation was observed. To address this issue, we introduce 4-(trifluoromethyl) phenylhydrazine (TFPH) to modify the perovskite solution, which presents enhanced storage stability. Consequently, when the modified solution was used to prepare PSCs, we obtained much improved and well consistent power conversion efficiencies (PCEs, ~ 26.0%) regardless of the perovskite solution ageing time, as well as exciting operational stability, which maintains PCE ≥ 92% for 1830 hours. These results are attributed to TFPH's multifunctionality: a) hydrazine groups inhibit perovskite decomposition by dual-pathway mechanism; b) trifluoromethyl boosts dipole moment, aiding crystallization and strain relaxation; c) impurity reduction and high-quality film jointly lower charge traps. This work substantially assists understanding and modifying perovskite degradation in both solution and solid phases. The developed performance stability and consistency on the TFPH modified device batches is of great significance for commercial production of PSCs.

Solution-processable metal halide perovskite solar cells (PSCs) have witnessed remarkable progress over the past decade, with power conversion efficiencies now approaching 27%[1]. This exceptional performance, combined with their low-temperature solution processability, makes PSCs a highly promising candidate for next-generation thin-film photovoltaics[2–4]. The solution-based fabrication method offers significant advantages, including low-cost production, compatibility with large-area and flexible substrates, and scalability for industrial manufacturing. However, the instability of perovskite precursor solutions remains a critical challenge, severely limiting the reproducibility and consistency of device performance during large-scale production[5,6]. Batch-to-batch variability induced by this

[1]Department of Materials Science and Engineering, Southern University of Science and Technology, Shenzhen, China. [2]Guangdong Provincial Key Laboratory of Energy Materials for Electric Power, Southern University of Science and Technology, Shenzhen, China. [3]SUSTech Energy Institute for Carbon Neutrality, Southern University of Science and Technology, Shenzhen, China. [4]Department of Materials Science and Engineering, City University of Hong Kong, Kowloon, China. [5]Future Photovoltaic Research Center, Global Institute of Future Technology, Shanghai Jiao Tong University, Shanghai, China. [6]Department of Chemistry, Southern University of Science and Technology, Shenzhen, China. [7]Urban Transformations Research Centre, Western Sydney University, Penrith, NSW, Australia. [8]These authors contributed equally: Guihua Zhang, Deng Wang, Bowei Li, Qing Lian. ✉e-mail: lianq@sustech.edu.cn; xubm@sustech.edu.cn; chengc@sustech.edu.cn

instability poses a significant challenge to the commercialization of PSCs[7]. As organic-inorganic hybrid materials, perovskites inherently exhibit relatively poor stability, with degradation occurring under both storage and operational conditions. The degradation of perovskite precursor solutions over time, influenced by many factors, including moisture, oxygen, and temperature, can result in substantial differences in the film formation, thereby introducing inconsistencies in the scale-up fabrication of PSCs[8]. These issues underscore the urgent need for strategies to enhance the stability of both perovskite solutions and solid-state films.

The chemical stability of perovskite precursor solutions is a fundamental prerequisite for the scalable manufacturing of high-performance PSCs. For instance, devices fabricated from aged solutions typically exhibit significantly lower power conversion efficiencies (PCEs) compared to those prepared using fresh solutions[7,9-11]. This primarily stems from the proton loss in organic salts, such as methylammonium ($MA^+$) and formamidinium ($FA^+$), and the oxidation of halides. Specifically, $MA^+$ readily undergoes deprotonation to form $MA^0$, which can further react with $FA^+$ to produce $MFA^+$, while the proton loss from FAI compromises solution stability even in the absence of $MA^{+12,13}$. These reactions disrupt the delicate chemical equilibrium in the precursor solution, leading to deviations from the stoichiometric ratio required for the formation of high-quality perovskite films and high-efficiency devices. Furthermore, the generated $I_2$ can disrupt the integrity of the perovskite lattice during crystallization, leading to the formation of defects, like iodide vacancies ($V_I$) and metallic lead ($Pb^0$)[14-16]. These defects accelerate perovskite degradation and damage adjacent functional layers, which collectively impede charge transport and overall device performance[17]. Recently, FA-based perovskite has emerged as a highly promising material for photovoltaic devices due to its broad light absorption spectrum and exceptional thermal stability[18-20]. However, the practical application of FA-based perovskite is also significantly hindered by stability issues. These degradation mechanisms pose critical challenges to the long-term performance and durability of perovskite solar cells. Given the pivotal role of formamidinium iodide (FAI) as a predominant precursor in FA-based perovskite systems, a comprehensive investigation into its ageing process is of paramount importance.

To address these challenges, significant efforts have been devoted to stabilizing perovskite solutions and films through additive engineering[9,21-23]. For example, hydrazine-based compounds, such as benzyl hydrazine hydrochloride (BHC), have been employed as additives to scavenge iodine species ($I_3^-$) from precursor solutions via redox reactions[11]. Lewis bases, such as thiourea and dimethyl sulfoxide (DMSO), have been employed to coordinate with undercoordinated $Pb^{2+}$ ions, thereby suppressing halide oxidation and improving solution stability[21,22]. Additionally, acidic additives, such as hydroiodic acid (HI) and hypophosphorous acid (HPA), have been introduced to mitigate the deprotonation of organic cations and maintain the chemical balance of the precursor solution[23]. Despite these advancements, the selection of effective additives remains challenging, as they must not only inhibit deprotonation and oxidation but also avoid interfering with the crystallization kinetics and morphology of the perovskite films. Achieving this balance is critical for ensuring both the stability of the precursor solution and the optoelectronic quality of the resulting perovskite films.

In this study, we conducted a systematic investigation into the ageing process of FA-rich perovskite solutions, revealing their severe spontaneous degradation when exposed to air. To mitigate this issue, we developed 4-(trifluoromethyl) phenylhydrazine (TFPH), a derivative of BHC[11], as a multifunctional additive to modify the perovskite precursor solution, which significantly enhanced the stability of perovskite in both solution and solid phases. TFPH acts through multiple complementary mechanisms: stabilizing the perovskite lattice via redox-active hydrazine moieties, guiding preferential crystal orientation through dipolar interactions, and minimizing trap density by simultaneously removing impurities and high-quality film. When the TFPH-modified solution was utilized for the fabrication of PSC devices, it consistently delivered high PCEs of approximately 26.0%, irrespective of the solution's ageing duration. Furthermore, the TFPH-modified PSCs exhibited markedly improved stability and performance consistency across device batches. Notably, the operational stability of these devices was significantly enhanced, retaining 92% of their initial efficiency after 1830 h under the ISOS-L-3 protocol.

## Results and discussion

Given that the organic-cation deprotonation and iodide oxidation are the main instability sources in this perovskite system[18], we first investigate the ageing process of the predominant perovskite precursors, namely formamidinium iodide (FAI). To accelerate the ageing process that happens in glove box, the FAI solution is exposed to the ambient atmosphere (22 °C, relative humidity (RH) ≥ 60%) for 3 days. In the following, if not specified, the ageing condition is the same as above. Figure 1a compares the $^1H$ NMR spectra of the fresh and the aged FAI solutions, where $c_1$ and $c_2$ are corresponded to -CH and $-NH_2$ groups in $FA^+$, respectively[22,24]. It is found that the integral intensity of $c_2$ peak ($A_{c2}$) drops from 4.03 (fresh) to 3.54 (aged), indicating an evident decrease in the concentration of $FA^+$. There is a sharp and strong peak newly appearing at 3.8 ppm (water peak), which indicates the presence of water that comes from air and from the $FA^+$ deprotonation. The color change provides more degradation information on the ageing of the FAI solutions. The fresh FAI solution is colorless while the aged one (3 days) exhibited a light-yellow color, indicating the generation of $I_2$. The absorption peak at 365 nm (Fig. 1b) from UV-Vis spectrum can be assigned as $I_3^-$ (the combination of $I_2$ and $I^-$) and this also confirms the presence of $I_2$[25,26]. The above results imply that the degradation of FAI leads to HI and the following oxidation product of $I_2$[13]. Similar experiments were carried out on the typical $FA_{0.95}Cs_{0.5}PbI_3$ perovskite solutions (Fig. 1c); this perovskite composition is commonly used for high-performance PSCs[27]. As the perovskite solutions are always light-yellow similar to $I_2$ solution, the observation by naked eyes and UV-Vis tests were conducted on their $I_2$ extracted solutions by toluene. It is hardly to observe light-yellow ($I_2$) color in all the toluene solutions by naked eyes (Fig. S1). However, from their UV-Vis spectra, a relatively weak $I_3^-$ peak appeared from the toluene solution, which was obtained from the aged FAI solution. From these above results, it is concluded that both FAI and the FA-rich perovskite solutions have the spontaneous degradation in air, which is resulted from halide oxidation and organic-cation deprotonation.

To mitigate the above degradation process, we introduce TFPH as a stabilizer to the studied FAI and perovskite solutions. In the presence of TFPH, the aged FAI solution maintains colorless, accompanied with the disappearance of the $I_2$ ($I_3^-$) absorption peak at 365 nm (Fig. 1b). These features were the same as those of the fresh FAI solution and thus it is concluded that TFPH can effectively inhibit FAI degradation[25,26]. Beside the FAI solution, TFPH also works well to stable the $FA_{0.95}Cs_{0.5}PbI_3$ perovskite solution. No absorption peak of $I_3^-$ was observed in the TFPH modified aged perovskite solutions. The above results indicate the common effect of TFPH in inhibiting the degradation of FAI-containing solutions.

To quantify the inhibiting effect of TFPH on the degradation of perovskite solution, we further characterize the above perovskite solutions aged in air for 1-3 days. As the ageing/storage time increases, the integral intensity of $c_2$ peak ($A_{c2}$) of the control solution ($FA_{0.95}Cs_{0.5}PbI_3$ solution without TFPH) rapidly decreases from 3.99 to 2.76 (Fig. 1d). For the target solution (the TFPH modified $FA_{0.95}Cs_{0.5}PbI_3$ solution), $A_{c2}$ only presents a slight decrease from 3.84 to 3.77 (Fig. 1e). The above results again confirmed the inhibiting effect of TFPH on the oxidation of $I^-$ to $I_2$ in the perovskite solution. Moreover, a fast increased water peak that shifts from 3.3 to 3.8 ppm with

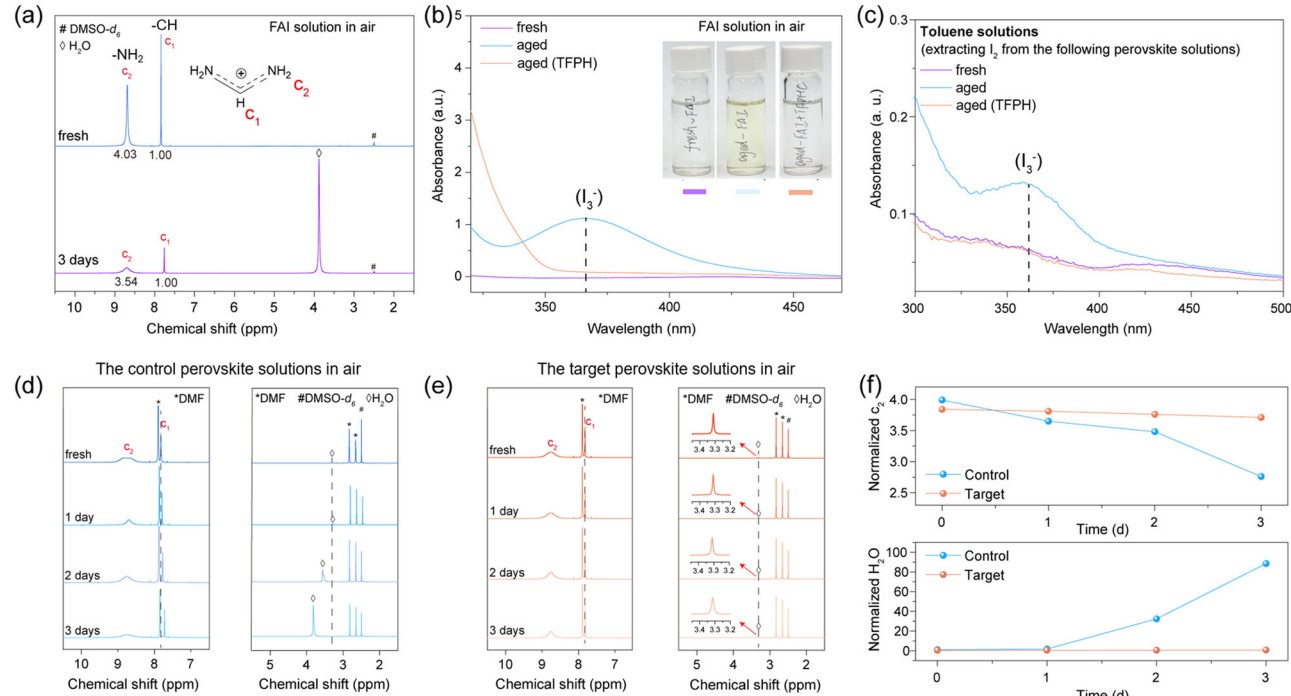

**Fig. 1 | Investigation of the ageing of FAI and perovskite solutions. a** [1]H NMR spectra and **b** the corresponding UV–Vis spectra of the fresh and aged FAI solutions in air. DMSO-d$_6$ is used for all [1]H NMR tests. The inset is the photograph of the FAI solutions in air. **c** UV–Vis spectra of perovskite solutions[1].H NMR spectra of the **d** control and **e** target perovskite solutions exposed to air for 1-3 days. **f** The percentage reduction of the integral area of c$_2$ and H$_2$O were obtained from the [1]H NMR spectra.

the ageing time was observed for the control solution while there is a rather weak and less-shifted water peak appearing for the target solution. As these solutions are aged in air, the rather weak peak intensity in the target solution indicates the solution hardly absorb water from the air. Moreover, the water peak is rather sensitive to the change of H$^+$ concentration and the above distinct changes of water peaks in the control perovskite solution, including its integral intensity (A$_w$) and peak position, reveals water in the control solution mainly comes from the degradation of perovskite. The weak and stable water peak reveals the effective inhibition of TFPH on the deprotonation of FA$^+$ to FA$^0$. The trends of two indicators (A$_{c2}$ and A$_w$) with the ageing time were re-plot in Fig. 1f for a better view. It is clear to show that as the ageing time increased, the target solution delivered minor change on the above two indicator values while the control one had dramatic changes, indicating that the introduced TFPH can effectively stabilize the perovskite solution.

The underlying mechanism TFPH inhibits the degradation of perovskite solution requires in-depth investigation. We thus mixed TFPH and I$_2$ and observed the resultant reactants, which can be revealed by the new peaks in both [1]H and [13]C NMR spectra. It is interesting to find that as the I$_2$/TFPH ratio increases from 0:1 to 10:1, the intermediate product of 4-trifluoromethylbenzoldiazonium[28] (TFBD, peaks located at 7.31 and 7.75 ppm) presented first and then gradually decreased while the final product of 4-iodotrifluorotoluene (TFBI, peaks located at 7.5 and 8.0 ppm) appeared and saturated (Fig. 2a, b). The identification of TFBD and TFBI can be referred to the Figs. S2 and S3 and the Mass spectrum of Fig. 2c (TFBI at 271.92 m/z).

According to Fig. 2b and Table S1, TFPH was consumed out by reacting with I$_2$ when the I$_2$/TFPH ratio reached 4:1 (Fig. 2b and Table S1). The amount of TFBD increased firstly and began to decline at the I$_2$/TFPH ratio of 3:1 while the amount of TFBI increased all along. The above results suggest that I$_2$ is preferred to react with TFPH to deliver the intermediate TFBD and the final product TFBI (Fig. 2c). The intermediate TFBD is transient and quickly transforms to TFBI, which

has a boiling point of 185–186 °C, comparable to that of DMSO (189 °C). Hence, TFBI is expected to be removed during the annealing process without negatively affecting the final perovskite film. Through the above reactions, I$_2$ is effectively reduced to I- and further trapped as TFBI if I$_2$ concentration further increases. Based on the results from Figs. 1 and 2a-c, the degradation and inhibition mechanisms of perovskite solution with TFPH are proposed as in Fig. 2d, e and reactions (1)–(5). The observation of N$_2$ gas release from the TFPH@I$_2$ DMF solution supports the above reactions (Fig. S4). Therefore, the main composition of perovskite, that is FAI, tends to degrade as I$_2$, H$_2$O, NH$_3$ and HCOOH by reacting with O$_2$ during the ageing process/storage (reactions (1)–(3)). These products, especially I$_2$, are reported to reduce the quality of perovskite film and deteriorate the PSC performance. Upon the introduction of TFPH, I$_2$ is reduced back to HI, inhibiting the deprotonation of FAI to FA$^0$ and HI (the reverse reaction (1)). It is demonstrated that hydrazine groups (–NH–NH$_2$) in BHC can reduce reactive iodine species (e.g., I$_2$) to iodide (I$^-$) via redox chemistry[11], thereby mitigating precursor solution degradation. However, our work reveals an unreported dual-pathway mechanism for I$_2$ suppression by the hydrazine group in TFPH. 1) Reduction Pathway at Low I$_2$ Concentrations: When I$_2$ levels are low (the I$_2$/TFPH ratio is less than 4:1), the hydrazine group (–NH–NH$_2$) in TFPH reduces I$_2$ to I$^-$, consistent with prior reports on hydrazine-based additives[9,11]. This mechanism was confirmed by NMR spectroscopy of TFPH/I$_2$ mixtures (Fig. 2a, b), with the reaction process illustrated in Fig. 2d (4) and 2e (4). Notably, we found that this reduction reaction occurs when I$_2$ concentrations are relatively low—not reported in the work with BHC as stabilizer[11]. 2) Nucleophilic Substitution Pathway at High I$_2$ Concentrations: Under excess I$_2$ conditions, TFPH directly scavenges I$_2$ through a nucleophilic substitution reaction, forming stable trifluoromethyl benzyl iodide (TFBI) and releasing inert N$_2$ gas (Fig. 2a, b, d (5), and e (5)). This pathway efficiently eliminates surplus I$_2$ without generating harmful byproducts, offering improved stabilization compared to conventional redox-based approaches. The released N$_2$ gas

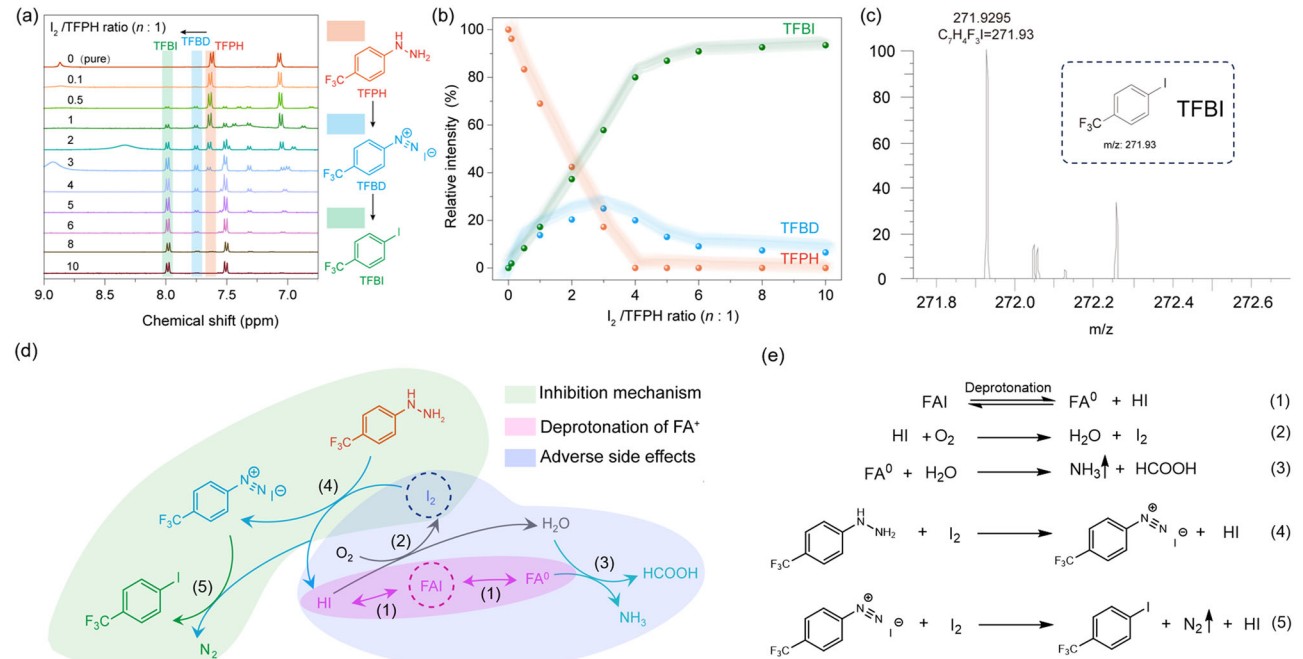

**Fig. 2 | Underlying mechanism of inhibiting solution ageing by reducing $I_2$. a** [1]H NMR spectra of TFPH and **b** ratio of [1]H NMR peak area at different $I_2$/TFPH ratios. **c** Mass spectrum of $I_2$/TFPH in MeOH. **d** Photographs of the reaction results between $I_2$ and TFPH. **e** Proposed reaction process of TFPH to inhibit the perovskite solution degradation.

further protects the perovskite from moisture and oxygen degradation (Fig. S4). This nucleophilic substitution pathway has not been disclosed or characterized in earlier studies[9,11]. As such, as storage station for $I_2$, TFPH and its intermediate TFBD can continuously accommodate $I_2$, thereby achieving "intelligent" regulation to keep the concentration of $I_2$ at a relatively low level. From above results, it is thus concluded that TFPH can not only inhibit the spontaneous degradation of perovskite, but also effectively eliminate harmful $I_2$ accumulated in the system, regardless of whether $I_2$ is preexisting or later produced.

From above, it is concluded that TFPH can effectively suppress the degradation of the perovskite solution. However, it is necessary, yet unknown, that how TFPH affects the quality of perovskite film. We thus comprehensively characterize and compared the perovskite films prepared with the fresh, the aged and the TFPH modified aged solutions (named as the control(fresh), control(aged) and target (TFPH modified and aged) films). It is found that the target film has the similar enlarged grain size as that of the control(fresh) film while the control(aged) film has a slightly reduced grain size, evidenced by both the top-view (Fig. S5) and cross-sectional SEM images (Fig. 3a). Notedly, TFPH can facilitate the oriented growth of perovskite film. The X-ray diffraction (XRD) patterns in Fig. 3b show that all samples exhibit a prominent diffraction peak at 13.9°, corresponding to the (001) crystallographic plane of the perovskite, which indicates preferred in-plane growth parallel to the substrate. A comparative analysis of the intensity ratio between the (001) and (111) peaks (Fig. 3c) reveals that the TFPH-treated (aged) film has a higher $I_{(001)}/I_{(111)}$ ratio than both the control (fresh) and control (aged) films. This demonstrates that TFPH preserves and enhances the preferential (001) orientation even after precursor ageing. Additionally, the full width at half maximum (FWHM) of the (001) peak decreases from 0.092° (control, fresh) to 0.087° (control, aged) and 0.084° (TFPH-treated, aged), indicating progressively improved crystallinity in the TFPH-treated film. These results suggest that TFPH promotes more ordered nucleation and growth, contributing to compact and highly oriented perovskite grains. Compared to the reported stabilizers that enhance the quality of perovskite films, including increasing grain size and reducing FWHM

of XRD diffraction peaks, our results rank among the best performers (Table S2), further demonstrating the effectiveness of the TFPH additive.

The grazing incidence X-ray diffraction measurements are performed to characterize the strain in the above perovskite films. As the penetration depth/tilt angle increases, the diffraction peak ($2\theta$) shifts to lower values from the initial position of 31.5° for all the studied films (Fig. S6). This behavior delivers negative fitted slopes of $2\theta$–$\sin^2\varphi$ (Fig. 3d), which indicates tensile strain in the films[29]. It is interesting to find that with the introduction of TFPH, the as-obtained perovskite film presents a decreased strain, which might be beneficial to improve device performance. Due to its large molecular size, TFPH is more likely to localize on the grain surface instead of merging into the perovskite lattice[24]. The decreased average roughness ($R_a$) obtained from atomic force microscopy (AFM, Fig. S7) and high-resolution X-ray photoemission spectroscopy (XPS) of C 1s and F 1s signals confirm the presence of TFPH on the perovskite surface (Fig. S8). The larger shifting binding energies of the Pb 4f and I 3d spectra (Fig. S9a, b) further confirm the stronger interaction between TFPH and perovskite. This enhanced interaction may be beneficial for reducing the notorious defects. All in all, the target film has large grain, high orientation, lower strain and strong surface TFPH molecular bonding; these merits may lead to decrease in overall defects concentration.

We further conducted the space-charge-limited current (SCLC) characterization to qualify the trap density of these perovskite films. According to the trap-filled limit voltages ($V_{TFL}$) (Fig. 3e), the trap density ($n_{trap}$) values are calculated as $1.03 \times 10^{15}$, $1.09 \times 10^{15}$, and $0.92 \times 10^{15}$ cm$^{-3}$, corresponding to the control(fresh), control(aged) and target films, respectively (Table S3). The lowest $n_{trap}$ of the target film is consistent with its high quality, even that it was prepared by the ageing solution. This result can be attributed to the two facts that 1) TFPH can significantly reduce the detrimental $I_2$ in the ageing solution and the as-prepared perovskite films as well; 2) TFPH can improve the crystallinity of perovskite film. As a consequent, the target film has a comparable UV-Vis absorption as that of the control(fresh) film

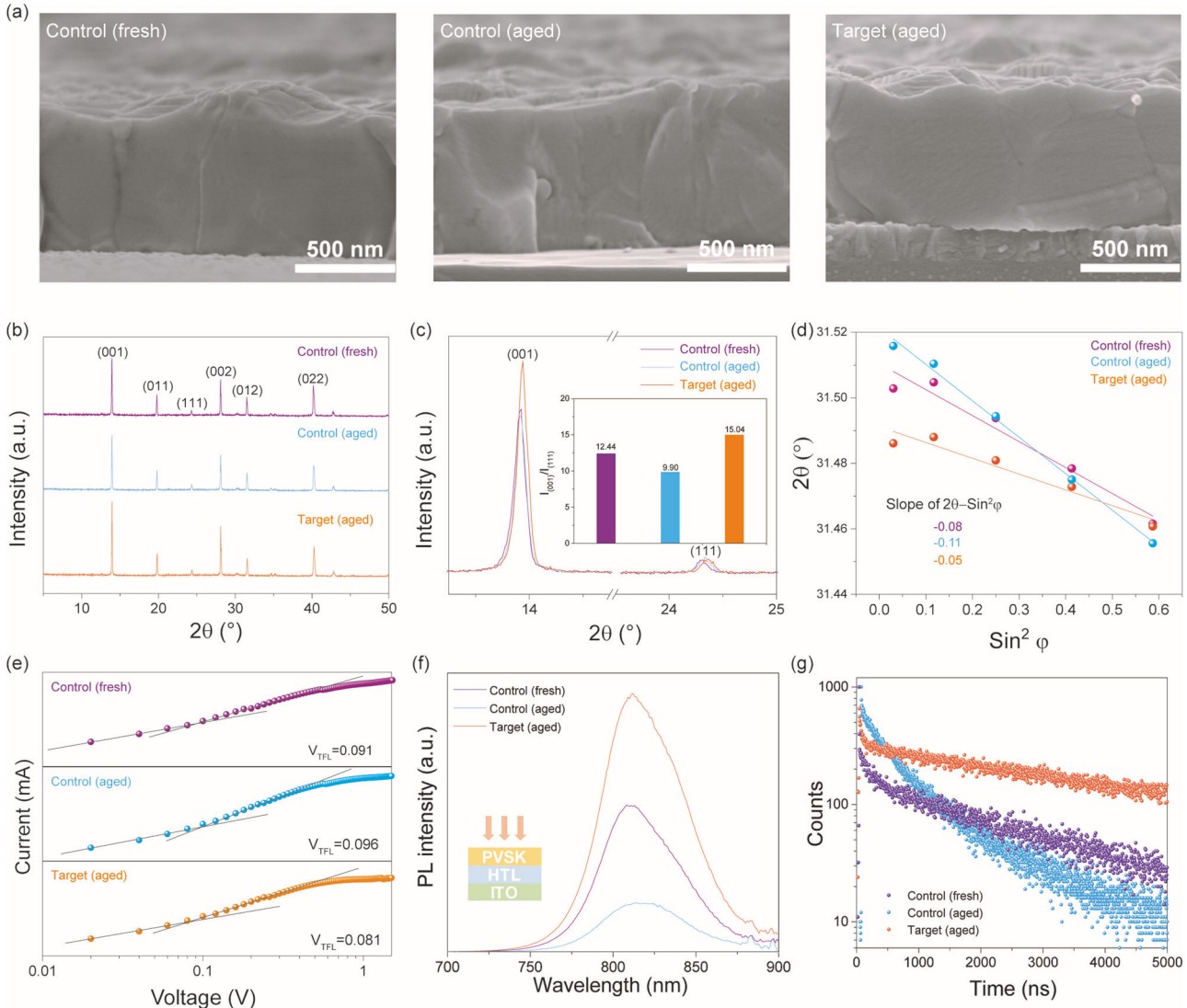

**Fig. 3 | Characterizations of the studied perovskite films. a** Cross-sectional SEM images. **b** XRD patterns. **c** Enlarged XRD areas (13.3–25°) with the calculated peak ratio of (001)/(111). **d** Strain analysis based on the 2θ-sin$^2$φ method. **e** Dark *I–V* curves of the hole-only devices. **f** Steady-state PL spectra. **g** Time-resolved PL decays.

(Fig. S9c) and exhibits a stronger photoluminescence (PL) emission and a longer PL lifetime (Fig. 3f, g, Table S4) than the two control films (fresh and aged). It is thus concluded that TFPH can improve the quality of perovskite film.

Based on the comprehensive characterization above, the mechanism by which TFPH stabilizes dual-phase perovskite and enhances the quality of perovskite films can be attributed to the synergistic integration of the hydrazine (-NH-NH$_2$) and trifluoromethyl (-CF$_3$) functional groups: The hydrazine group reduces I$_2$ to I$^-$ at low concentrations while directly incorporating I$_2$ into TFBI at high concentrations (Fig. 2). The released byproduct, N$_2$ gas, further shields the perovskite from moisture and oxygen-induced degradation (Fig. S4). Meanwhile, the -CF$_3$ group improves thin-film quality through multiple mechanisms: (1) dipole-modulated crystallization, which promotes preferential orientation and facilitates strain relaxation (Fig. S10); (2) enhanced wetting of perovskite precursors on SAMs-modified ITO surfaces (Fig. S11); and (3) effective defect passivation coupled with increased hydrophobicity (Fig. S12). The synergistic effects of these two functional groups in TFPH not only benefit the initial performance on both efficiency and stability of the fabricated PSCs but also ensure notable batch-to-batch reproducibility in device performance, as demonstrated below.

It is anticipated to obtain high-performance PSCs with high-quality target perovskite films. For this purpose, we fabricated a series of devices based on the control and the target perovskite solutions at different ageing time. The device structure is illustrated in Fig. 4a, b, c shows the champion PCEs of the control and the target devices fabricated with the fresh perovskite solutions and the aged perovskite solutions. Compared to the control device with a PCE of 23.7%, the target device has a greatly enhanced PCE of 25.9%, which is achieved by the remarkable improvements on $V_{oc}$ (1.12 to 1.19 V) and FF (0.83 to 0.86). The satisfactory $V_{oc}$ results from the improved crystallization and reduced trap density; the elevated FF from the compact and highly oriented perovskite grains and smooth surface, favoring the carrier transportation. Therefore, the above result is consistent with the better quality of the target perovskite film that is modified by TFPH (Fig. 3). Figure 4d and Table 1 show the performance change of the control and the target batched devices with the ageing time of the perovskite solutions increases (up to 60 days). The PCE of the target device remains almost unchanged after 60 days of ageing (only a 0.04% decrease), with a standard deviation consistently between ±0.29% and ±0.33%, indicating stable and uniform performance. In contrast, the PCE of the control device decreases significantly (6.17% drop), and its standard deviation increases over time (from ±0.40% to

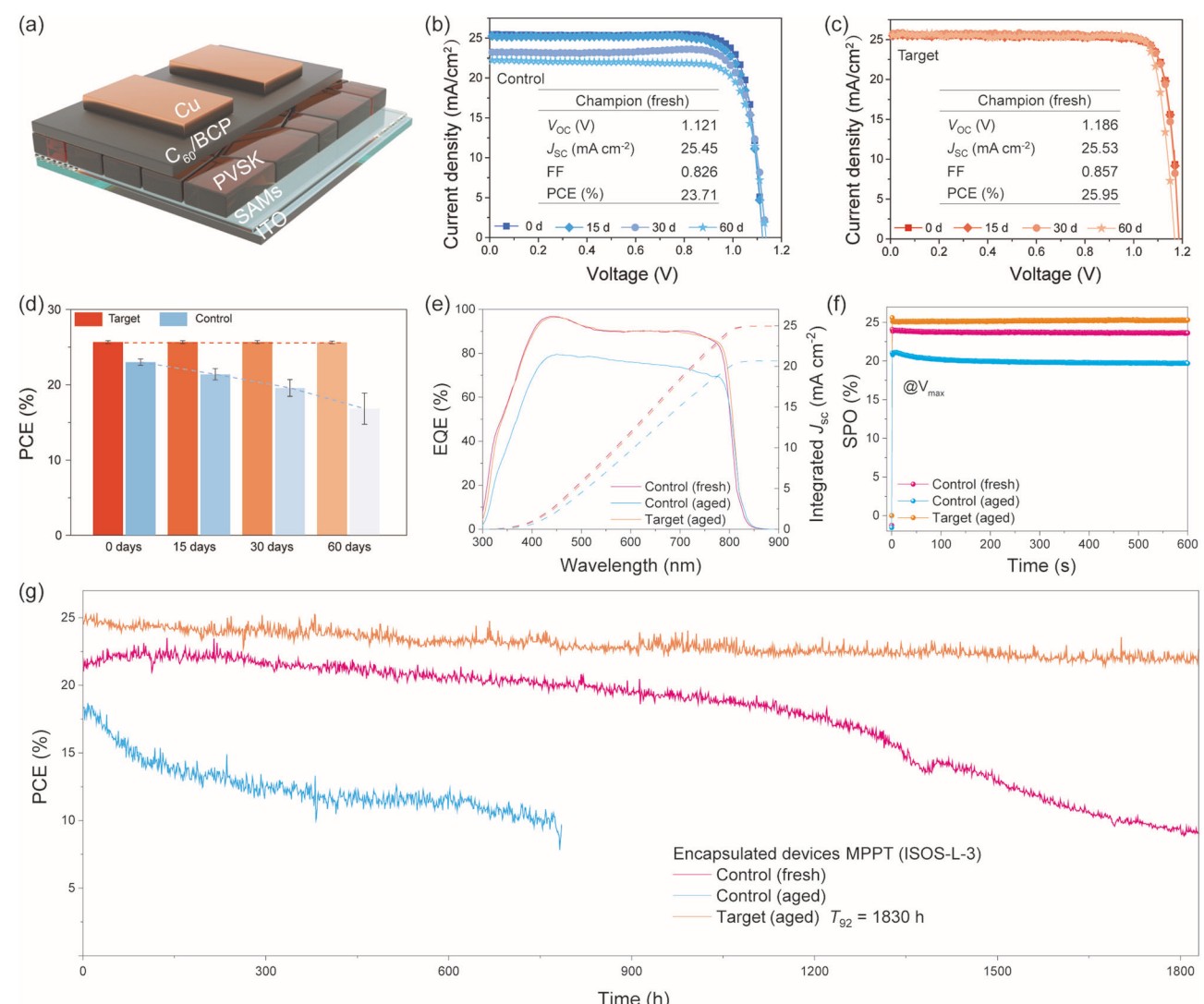

**Fig. 4 | Device performance with or without TFPH. a** Schematic structure of the fabricated devices. **b, c** J–V curves of perovskite solution at various ageing times. **d** Statistic data of perovskite solution, stored in a glovebox, at various ageing times. **e** Corresponding EQE spectra. Note that the aged results were obtained from the devices based on the solution after 30-day ageing. **f** Stabilized power output (SPO) at a fixed bias of $V_{max}$. **g** Long-term operational stability under maximum power point tracking (MPPT) is determined by the perturb and observe method.

**Table 1 | PCEs of the batched PSCs based on the control and target solutions**

| | | 1 day | 15 days | 30 days | 60 days |
|---|---|---|---|---|---|
| Control | Average | 22.99 ± 0.40% | 21.39 ± 0.74% | 19.58 ± 1.12% | 16.82 ± 2.07% |
| | Champion | 23.71% | 22.70% | 21.74% | 20.21% |
| Target | Average | 25.66 ± 0.29% | 25.67 ± 0.31% | 25.67 ± 0.33% | 25.62 ± 0.29% |
| | Champion | 25.95% | 25.98% | 26.00% | 25.91% |

±2.07%), indicating poor stability and consistency. The observed performance degradation is primarily attributed to the significant reduction in $J_{SC}$ with ageing time. Specifically, the control devices exhibited a substantial decrease in $J_{SC}$ from 25.38 to 20.68 mA cm$^{-2}$, accompanied by an increase in standard deviation from 0.42 to 1.16. In marked contrast, the target devices demonstrated better stability, maintaining $J_{SC}$ values above 25.5 mA cm$^{-2}$ throughout the ageing process with minimal variation, indicating notable reproducibility. This performance stability can be ascribed to the well-preserved precursor composition stabilized by TFPH. Thus, the introduction of TFPH in perovskite solution can effectively improve the stability and

consistency of device batches, which is of great significance for commercial device production. The data statistics in Fig. 4d is listed in Table S5, S6 for the convenience of readers. Moreover, it is found that the PCE degradation observed in the control device with the aged time comes from the sharp decline in $J_{SC}$, which is attributed to the accumulation of $I_2$ in the perovskite solution with the ageing time. We calculated the integrated $J_{SC}$ from external quantum efficiency (EQE) spectra where the control (fresh), control (aged 60 days) and target (aged 60 days) devices show 24.94, 20.67 and 24.93 mA cm$^{-2}$, respectively (Fig. 4e), with a negligible mismatch between the J-V and EQE measurements. Figure 4f shows the steady-state power output (SPO) at

the maximum power point voltage ($V_{MPP}$). The control (fresh) and target (aged 60 days) devices demonstrated rather stable SPO of 23.7% and 25.6%, respectively, while the control (aged 60 days) device exhibited a gradually decreased SPO (initial value 21.1%) over time. These results suggest the introduction of TFPH is also helpful to inhibit the perovskite degradation at the operation conditions and thus greatly enhance the operation stability of the PSCs.

We further compared device performance under harsher conditions, specifically ageing in ambient air (Fig. S13a). Devices were fabricated from perovskite precursors with and without TFPH, using both fresh and air-aged solutions over several days. It is found that the stability advantage persists under prolonged aging: after 2, 4, and 8 days, the target devices maintain PCEs of 24.91%, 23.88%, and 22.51%, respectively, whereas the control devices degrade rapidly to 20.71%, 18.32%, and 15.77%. These results robustly demonstrate that TFPH incorporation effectively mitigates precursor degradation, even under humid air conditions (Fig. S13b and Table S8 and S9).

To verify the above deduction, we conducted long-term stability assessments following the ISOS-L-3 testing protocols. The encapsulated PSCs were illuminated under continuous 1-Sun intensity at the maximum power point (MPP) at 65 °C and 60% RH. As shown in Fig. 4g, the target (aged 60 days) device still maintained 92% of its initial PCE ($T_{92}$) after 1830 h, which is one of the best lifetimes in ISOS-L-3 assessments so far (Table S10). In contrast, both the control (fresh) and the control (aged 60 days) devices present bad operation stability with fast decay on PCE. The above results are quite interesting as they suggest that TFPH can inhibit the degradation of perovskite film under the harsh operation conditions. To gain an in-depth understanding of the additive's stabilizing effect, we systematically characterized the control and target perovskite films before and after 200 h of light soaking. The comprehensive analyses (Figs. S14–S16) consistently confirm that TFPH effectively stabilizes the target perovskite films, in line with the observed notable operational stability. This sustained operation stability of target devices is achieved by the firmly bonded TFPH molecules on the surface of perovskite grains with the same mechanism that TFPH works in the aged perovskite solution as manifested in Fig. 2.

Furthermore, we successfully incorporated TFPH into both narrow-bandgap (1.25 eV) and wide-bandgap (1.75 eV) perovskites, observing consistent performance enhancements in the corresponding devices, as confirmed by their improved current density-voltage ($J$–$V$) characteristics (Fig. S17). These results highlight the universal efficacy of TFPH as a versatile additive, suggesting its broad applicability in optimizing perovskite solar cells (PSCs) with different bandgaps—an important step toward their commercialization. To further evaluate the practical viability of TFPH, we performed a detailed cost-benefit analysis (Tables S11 and S12), which revealed that the modified perovskite precursor incurs only a minimal cost increase of 0.005 USD per mL while achieving a substantial 11.62% improvement in power conversion efficiency (PCE). This compelling cost-to-performance ratio underscores both the industrial feasibility and economic advantage of TFPH integration in PSC fabrication.

In summary, this work reveals that oxidative degradation of FA-rich perovskite precursors in air—particularly $I_2$ generation—severely compromises film quality and device performance. By introducing TFPH as a multifunctional stabilizer, we simultaneously suppress solution-phase decomposition and elevate the quality of resulting perovskite films. The optimized devices achieve consistent PCEs of ~26.0% with satisfactory operational stability, overcoming batch-to-batch variability. The universal effectiveness of this approach is demonstrated through its successful application to normal, narrow- and wide-bandgap perovskite compositions, establishing TFPH's ability to mitigate moisture- and oxygen-induced degradation pathways. Beyond its standalone effectiveness, TFPH's molecular design suggests strong potential for synergistic combinations with other stabilizers—

such as polymer additives or interfacial modifiers—to further enhance device performance and longevity. Our work provides both fundamental insights into perovskite aging mechanisms and a practical, scalable strategy for industrial PSC production.

## Method

### Materials
Formamidinium iodide (FAI, 99.9%), cesium iodide (CsI, 99.999%), lead iodide ($PbI_2$, 99.999%), and fullerene ($C_{60}$, 99.9%) were purchased from Ying Kou You Xuan Trade Co., Ltd. Bathocuproine (BCP, 99.99%) was obtained from Xi'an Polymer Light Technology Corp. 4-(tri-fluoromethyl)-phenylhydrazine hydrochloride (TFPHC, 97%) was purchased from J&K Scientific. 2PADCB was synthesized in the lab according to the previous methodology in the literature. N, N-dimethylformamide (DMF, anhydrous, 99.8%), dimethyl sulfoxide (DMSO, anhydrous, 99.9%), 2-propanol (IPA, anhydrous, 99.5%), chlorobenzene (CB, anhydrous, 99.8%), dichloromethane (DCM, anhydrous, 99.9%) and methanol (MeOH, anhydrous, 99.8%) were obtained from Sigma–Aldrich. All materials were used as received.

### Preparation of TFPHPbCl₃ crystals
$TFPHPbCl_3$ crystals were grown with the slow vapor diffusion of anti-solvent DCM into the precursor solution. The precursor solution was prepared by dissolving 425 mg of TFPHC (2 mmol) and 556 mg of $PbCl_2$ (2 mmol) in 2 mL of the DMSO solvent. Then, the precursor solution was filtered with a 0.22 μm PTFE membrane filter to obtain a clear solution. The glass vial with clear solution was exposed to a 20 mL vial filled with 10 mL of dichloromethane and sealed with parafilm at room temperature for 30 days. Next, the solids were collected and washed with dichloromethane. Finally, white crystals were vacuum-dried for 24 h. The thus pretreated additives are hereafter referred to as TFPH.

### Perovskite precursor solution
For the composition $FA_{0.95}Cs_{0.05}PbI_3$, 1.8 M perovskite precursor solution was prepared by dissolving 2.937 g FAI, 234 mg CsI, and 8.381 g $PbI_2$ in 10 mL mixed solvents of DMF and DMSO (v/v, 4/1) under continuous overnight stirring. For the target precursor solution, 2.937 g FAI, 234 mg CsI, 8.381 g $PbI_2$ and 5 mg of TFPH were dissolved in 10 mL mixed solvents of DMF and DMSO (v/v, 4/1) with continuous overnight stirring. Then, the precursor solution was filtered by a 0.22 μm PTFE filter before usage. The perovskite precursors for device fabrication were aged in an $N_2$-filled glovebox with oxygen levels <10 ppm and water levels <0.1 ppm.

### Device fabrication
The planar p-i-n perovskite solar cells were fabricated with an architecture of ITO/2PADCB/$Cs_{0.05}FA_{0.95}PbI_3$/$C_{60}$/BCP/Cu. Patterned ITO glass was ultrasonically cleaned for 20 min with a detergent, deionized water, acetone and ethanol, sequentially. Then, the ITO-coated glass substrates were treated with ultraviolet ozone for 20 min after being dried by an $N_2$ gun, then the treated ITO glass was transferred to the glove box. 2PADCB solution (0.5 mg mL$^{-1}$) in MeOH was spun onto the above ITO substrate at 3000 rpm for 5 s, and then annealed at 100 °C for 5 min. The perovskite precursor solution was spread on the 2PADCB-coated substrate at 2000 rpm for 10 s and then at 5000 rpm for 30 s. Then, 130 μL of CB was quickly dropped in 10 s before the end of the procedure. The wet films were immediately transferred to a heating plate and annealed at 100 °C for 30 min. A saturated PDI solution was spin-coated on the perovskite surface at 5000 rpm and annealed 100 °C for 5 min as post-treatment. Devices were completed by the thermal evaporation of 40 nm $C_{60}$, 8 nm BCP, and 100 nm Cu.

### Device characterization
The current density–voltage ($J$–$V$) curves and steady-state output (SPO) were measured with an active area of 0.085 cm$^2$ mask by a

Keithley 2400 source measurement unit and a solar simulator with an AM1.5G spectrum. The light intensity was calibrated by Newport calibrated reference Si solar cell. EQE spectra were measured under ambient conditions by a QE-R 3011 system with 210 Hz chopped monochromatic light ranging from 300 to 900 nm. For the stability measurements, encapsulated devices were tracked at MPP on a 65 °C hotplate in 60% relative humidity under 1 Sun illumination, corresponding to the ISOS-L-3 protocol.

## Other characterizations

UV–Vis absorption spectra were obtained using a PerkinElmer Lambda 950 UV–Vis spectrometer. For the precursor solution, toluene was used to extract iodine from the precursor solution. For the FAI-DMF solution, 1.8 M FAI with/without TFPH was dissolved in DMF and left in air at room temperature for aging. NMR spectra were acquired by using a Bruker AVANCE-III 400 MHz spectrometer. XPS measurements were conducted on a Thermo Scientific K-Alpha XPS system with a monochromatic aluminum (K$\alpha$) X-ray source providing photons with the energy of 1486.7 eV. XRD measurements were done by using a Rigaku Smart Lab 3 kW X-ray diffractometer with a Cu (K$\alpha$, 1.5406 Å) source at 10° min$^{-1}$ scan rate and $2\theta$ range of 5–50°. The top-view and the cross-sectional images of the films were obtained by a field-emission SEM (Regulus 8230). Steady-state PL (excitation at 405 nm) and TRPL spectroscopy were performed using an Edinburgh FLS1000 spectrometer. SCLC measurements were performed on the hole-only device stacks using a Keithley 4200 system under dark conditions. MS spectra were acquired from a Q Exactive Orbitrap Mass Spectrometer. The AFM images of surface morphology were conducted by Bruker Multimode 8 in air (RH 40–50%, RT 25 °C). The dipole moment of phenylhydrazine and 4-(trifluoromethyl)-phenylhydrazine was calculated based on the density functional theory using the Gaussian 09 software package. The structure optimization was carried out at B3LYP/6-311 g, and the calculated results were analyzed by GaussView.

## Reporting summary

Further information on research design is available in the Nature Portfolio Reporting Summary linked to this article.

## Data availability

The main data supporting the findings of this study are available within the published article and its Supplementary Information and source data files. Additional data are available from the corresponding author on request. Source data are provided with this paper.

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

## Acknowledgements

This work was supported by the National Key Research and Development Project funding from the Ministry of Science and Technology of China (Grant No. 2021YFB3800101, B.X.), Basic Research Project of Science and Technology Plan of Shenzhen (Grant No. 20231115112954001, C.C.), the National Natural Science Foundation of China (Grant No.

22305111 (Q. L.), 22409130 (B.L.) and 52173171 (X.G.)), Guangdong-Hongkong-Macao Joint Laboratory (No. 2019B121205001, C.C.), Guangdong Provincial Key Laboratory of Energy Materials for Electric Power (No. 2018B030322001, C.C.), High level of special funds (G03034K001, C.C.), The Shanghai Science and Technology Innovation Action Plan (24DZ3001203, B.L.). The authors also acknowledge the supports received from Fundamental Research Funds for the Student Innovation Training Program (Grant Nos. 2022G01, 2022G02, 2023S03, 2023×01, 2023×02, 2023×03, C. C.), Southern University of Science and Technology (SUSTech), and special funds for Cultivation of Guangdong College Students' Scientific and Technological Innovation (Grant Nos. pdjh2022c0005, pdjh2023b0460 and pdjh2024c10910, C.C.). The authors acknowledge H. Yi at the Southern University of Science and Technology and SUSTech Core Research Facilities for assistance in the characterization of perovskite films.

## Author contributions

G.Z., Q.L. and C.C. conceptualized the work. G.Z., D.W. and Q.L. fabricated and characterized solar cells. B.L., Q.L., G.Z. and C.C. wrote the manuscript. G.Z., X.Z. and K.F. conducted NMR measurement. D.L., Q.Y., G.M., J.L. and A.A. carried out film measurements and analyzed the data. Q.L., B.L., X.G. and C.C. acquired funding. Q.L., B.L., A.J., X.G., B.X. and C.C. supervised this work. All authors participated in revising and proofreading the final manuscript.

## Competing interests

The authors declare no competing interests.
