## [Transparent Peer Review file · Nature Communications]

Stabilizing Dual-Phased Perovskite Towards High Performance Photovoltaics with Enhanced Batch Stability and Consistency

Corresponding Author: Professor Chun Cheng

Version 0:

Reviewer comments:

Reviewer #1

(Remarks to the Author)

This manuscript presents a significant advancement in the field of perovskite solar cells (PSCs) by addressing the critical issue of solution and solid-phase stability, which is a major bottleneck for the commercialization of PSCs. The introduction of 4-(trifluoromethyl)phenylhydrazine (TFPH) as a stabilizer not only enhances the storage stability of perovskite solutions but also improves the performance consistency and operational stability of the resulting devices. The work provides a deep understanding of the degradation mechanisms and offers a practical solution to improve batch-to-batch reproducibility. The findings are well-supported by the experimental data and provide a clear pathway for improving the stability and reproducibility of PSCs. I recommend acceptance for publication after some revisions to further enhance the clarity and impact of the manuscript.

1. The introduction could benefit from a more detailed discussion on the current challenges in perovskite solution stability, particularly focusing on the impact of batch-to-batch variability on industrial scalability. This would strengthen the motivation for the study.
2. It would be helpful to include a brief discussion on how the degradation products (e.g., I₂) affect the film morphology and device performance. This would provide a more comprehensive understanding of the problem.
3. The labels for the intermediate and final products (TFBD and TFBI) in Figure 2 should be more prominent. Additionally, a brief discussion on how these products contribute to stabilizing the perovskite solution would be useful.
4. It is beneficial to include a comparison with other stabilizers reported in the literature, particularly in the discussion of improved film quality (e.g., grain size, crystallinity), to highlight the uniqueness of TFPH.
5. It is better to include a comparison of films' and devices' properties (TOF-SIMS, XRD, etc) before and after long-term sunlight stability tests to highlight and provide more insights into their unique TFPH.

Reviewer #2

(Remarks to the Author)

The authors report a stabilization strategy of perovskite precursor by incorporating a multifunctional molecule of TFPH. They first reveal the precursor degradation process in air, and then show the beneficial roles of TFPH in inhibiting solution degradation and improving the film quality of perovskite films. They finally demonstrated devices with efficiencies of around 26% and decent stability by utilizing solutions after storage in the glovebox for two months.

The proposed strategy of using TFPH additive is effective in inhibiting the precursor degradation, however, the reviewer found no new or advanced understanding of the precursor degradation mechanism compared with that demonstrated in previous reports (Science Advance, 2021 7, eabe8130). In addition, the mechanism of TFPH-inhibited precursor degradation seems also no obvious difference with that reported in previous work (Science Advance, 2021 7, eabe8130). As two key findings in the present work show no obvious advancements, the reviewer considers the manuscript not suitable for publication in Nature Communications.

Some detailed comments are listed below:

1. The authors characterized molecular interactions between the TFPH molecule and I₂ to understand the mechanism of the inhibited precursor degradation. It is clear that the TFPH possesses two key functional moieties of hydrazine and trifluoromethyl; the authors need to separately discuss the role of the two functional groups in stabilizing the precursor

solution. In addition, previous work (Science Advance, 2021 7, eabe8130) using similar additives of benzyl/propyl/(2-thienylmethyl) hydrazine hydrochloride has confirmed the role of hydrazine groups in stabilizing the precursor solution, is there any new mechanism for the TFPH in the present work? These questions, which are important for clarifying the rationale of the work, must be properly addressed.

2. In Figure 1 d and e, the 'Intensity (a.u.)' for the y-axis of the NMR results seems unnecessary.

3. The authors presented device results utilizing precursors with different aging time in the glovebox, it would be necessary to present the performance evolution for the two types of precursors with different aging time in ambient air?

Reviewer #3

(Remarks to the Author)

This manuscript presents a highly significant and timely study on the stabilization of perovskite solar cells (PSCs) using 4-(trifluoromethyl)phenylhydrazine (TFPH) as a solution-phase and solid-phase stabilizer. The work addresses a critical challenge in the field of perovskite photovoltaics—namely, the instability of perovskite precursor solutions and films—which is a major barrier to the commercialization of PSCs. The authors demonstrate a novel and effective approach to enhance both the storage stability of perovskite solutions and the operational stability of devices, achieving remarkable power conversion efficiencies (PCEs) of ~26.0% with excellent batch-to-batch consistency. The study is well-executed, with a strong emphasis on understanding the underlying mechanisms of TFPH's stabilizing effects, making it a valuable contribution to the field. I recommend this work for publication after addressing the following issues:

1. The abstract provides a good overview of the study, but it could benefit from a more detailed explanation of the specific mechanisms by which TFPH stabilizes the perovskite solution and enhances device performance. For instance, the abstract briefly mentions that TFPH inhibits degradation, promotes oriented crystallization, and reduces trap density, but it does not elaborate on how these effects are achieved. Including a few more sentences to explain these mechanisms would make the abstract more informative and compelling. Additionally, the abstract could highlight the broader implications of this work for the commercialization of PSCs.

2. The introduction is well-structured and provides a solid background on the challenges of perovskite stability. However, it could benefit from a more detailed discussion of the current state of perovskite solar cell commercialization and how this study addresses specific industrial challenges. For example, the introduction could include a brief overview of the current limitations in large-scale production and how TFPH offers a solution to these problems. This would help readers better understand the significance of the work in the context of real-world applications.

3. The manuscript lacks detailed experimental procedures, particularly in the preparation of perovskite solutions and device fabrication. Adding a more comprehensive methods section would improve reproducibility and allow other researchers to replicate the study. For example, the authors should provide specific details on the concentrations of TFPH used, the exact aging conditions, and the steps involved in device fabrication. This would enhance the transparency and reliability of the study.

4. While the PCE is highlighted, other important metrics such as fill factor (FF), open-circuit voltage (Voc), and short-circuit current (Jsc) should be discussed in more details. For instance, the authors could provide a more comprehensive analysis of how TFPH affects these parameters and how they contribute to the overall device performance. This would provide a more complete picture of the device's performance.

5. The XRD patterns are mentioned, but a more detailed discussion of the crystallographic changes induced by TFPH would be beneficial. For example, the authors could provide a more in-depth analysis of the XRD data, including the crystal structure and orientation of the perovskite films. This would help readers understand how TFPH affects the crystallization process.

6. A cost analysis of using TFPH in large-scale production would provide valuable insights into its commercial viability. For example, the authors could estimate the cost of TFPH synthesis and its impact on the overall cost of PSC production. This would help readers assess the economic feasibility of using TFPH in commercial devices.

7. The conclusion is concise but could be expanded to summarize the key findings and their implications for the field of perovskite solar cells more comprehensively. For example, the authors could discuss how their findings could be applied to other types of perovskite devices or how TFPH could be used in combination with other stabilizers to further improve performance. This would provide a more forward-looking perspective on the study.

Version 1:

Reviewer comments:

Reviewer #1

(Remarks to the Author)

The revised manuscript can now be accepted.

Reviewer #2

(Remarks to the Author)

My key concern was with the new mechanism/understanding of stabilizing the precursor by using the presented TFPH and reported BHC featuring a similar functional moiety and reaction path. In the revised manuscript, the authors provided a detailed comparison and interpretation of the different aspects of TFPH, demonstrating the advantages of the TFPH and making the mechanistic understanding clear. As the BHC is functional similarly on stabilizing the precursor solution, it would be more appropriate to include the work in the third paragraph of the introduction. In addition, the key difference on the new mechanism compared with the BHC should also be included in the discussion part, which would be respectful to previous work. With these changes, the work would be considerable for publication.

Reviewer #3

(Remarks to the Author)

The authors have revised this manuscript following reviewers' comments, it can be accepted as the present form.

Response Letter to Reviewers

We sincerely thank the reviewers for their constructive comments on our manuscript. Below, we provide detailed responses to each point. All revisions are highlighted in the updated version of the manuscript.

Response to Reviewer #1:

This manuscript presents a significant advancement in the field of perovskite solar cells (PSCs) by addressing the critical issue of solution and solid-phase stability, which is a major bottleneck for the commercialization of PSCs. The introduction of 4-(trifluoromethyl) phenyl hydrazine (TFPH) as a stabilizer not only enhances the storage stability of perovskite solutions but also improves the performance consistency and operational stability of the resulting devices. The work provides a deep understanding of the degradation mechanisms and offers a practical solution to improve batch-to-batch reproducibility. The findings are well-supported by the experimental data and provide a clear pathway for improving the stability and reproducibility of PSCs. I recommend acceptance for publication after some revisions to further enhance the clarity and impact of the manuscript.

Response: We highly appreciate the reviewer for highlighting the importance of this study. We have addressed the specific concerns raised by the reviewer below.

Q1: The introduction could benefit from a more detailed discussion on the current challenges in perovskite solution stability, particularly focusing on the impact of batch-to-batch variability on industrial scalability. This would strengthen the motivation for the study.

Response: Thank you for your suggestion. We have added more detailed discussion in the Introduction to strengthen the motivation behind our study:

“Batch-to-batch variability induced by this instability poses a significant challenge to the commercialization of PSCs (*ACS Materials Letters* **2021**, 3, 351). As organic-inorganic hybrid materials, perovskites inherently exhibit relatively poor stability, with degradation occurring under both storage and operational conditions. The degradation of perovskite precursor solutions over time, influenced by many factors, including moisture, oxygen, and

temperature, can result in substantial differences in the film formation, thereby introducing inconsistencies in the scale-up fabrication of PSCs (*Nature Communications* **2024**, *15*, 4552). These issues underscore the urgent need for strategies to enhance the stability of both perovskite solutions and solid-state films.”

We have updated the manuscript accordingly at page 3, lines 11-19.

Q2: It would be helpful to include a brief discussion on how the degradation products (e.g., I₂) affect the film morphology and device performance. This would provide a more comprehensive understanding of the problem.

Response: Thanks for the reviewer’s constructive suggestion. We have added relative discussion in the revised Introduction outlining the impact of molecular iodine to the film morphology and device performance.

“Furthermore, the generated I₂ can disrupt the integrity of the perovskite lattice during crystallization, leading to the formation of defects, like iodide vacancies (V_I) and metallic lead (Pb⁰) (*Nature Materials* **2024**, *23*, 810; *Energy & Environmental Science* **2024** *16*, 6071; *Advanced Functional Materials* **2024**, *35*, 2414423). These defects accelerate perovskite degradation and damage adjacent functional layers, which collectively impede charge transport and overall device performance (*Small* **2025**, *21*, 2410776).”

We have updated the manuscript accordingly at page 4, lines 1-5.

Q3: The labels for the intermediate and final products (TFBD and TFBI) in Figure 2 should be more prominent. Additionally, a brief discussion on how these products contribute to stabilizing the perovskite solution would be useful.

Response: Thanks for the reviewer’s constructive suggestion. We have enlarged the labels and improved the quality of **Figure 2** for a clear view, as shown in **Figure R1**. According to the reaction pathway shown in **Figure 2**, the only non-gaseous products are TFBD and TFBI. NMR results (**Figure 2a** and **2b**) reveal that the intermediate TFBD is transient and will quickly transform to TFBI. The main composition of perovskite in solution, that is FAI, tends to degrade as I₂, H₂O, NH₃ and HCOOH by reacting with O₂ during the ageing process/storage (reactions (1)-(3)). These products, especially I₂, are reported to reduce the quality of perovskite film and deteriorate the PSC performance. Upon the introduction of

TFPH, I_2 is reduced back to HI, inhibiting the deprotonation of FAI to FA^0 and HI (the reverse reaction (1)).

When I_2 is accumulated, TFPH can reduce I_2 through reaction (4) and the resulted TFBD further eliminates I_2 through reactions (5) at relatively high concentration of I_2 , with TFBI as the final product. As a storage station for I_2 , TFBI can accommodate more I_2 , thereby achieving "intelligent" regulation to keep the concentration of I_2 at a low level. Above all, these results show that TFPH can not only inhibit the spontaneous degradation of perovskite, but also effectively eliminate harmful I_2 accumulated in the system, regardless of whether I_2 is preexisting or later produced. In additional, since TFBI has a boiling point of 185-186 °C, comparable to that of DMSO (189 °C), it is expected to be removed with perovskite solvent during the annealing process. We have added the corresponding discussion in the revised manuscript:

“The intermediate TFBD is transient and quickly transforms to TFBI, which has a boiling point of 185-186 °C, comparable to that of DMSO (189 °C). Hence, TFBI is expected to be removed during the annealing process without negatively affecting the final perovskite film.” (Page 8, lines 6-10)

“When I_2 is accumulated, TFPH can reduce I_2 through reaction (4) and the resulted TFBD further eliminates I_2 through reaction (5) at relatively high concentration of I_2 , with TFBI as the final product. As such, as storage station for I_2 , TFPH and its intermediate TFBD can continuously accommodate I_2 , thereby achieving "intelligent" regulation to keep the concentration of I_2 at a relatively low level.” (Page 8, lines 20-24)

Figure R1 (Figure 2). Underlying mechanism of inhibiting solution ageing by reducing I₂. a) ¹H NMR spectra of TFPH and b) ratio of ¹H NMR peak area at different I₂/TFPH ratios. c) Mass spectrum of I₂/TFPH in MeOH. d) Photographs of the reaction results between I₂ and TFPH. e) Proposed reaction process of TFPH to inhibit the perovskite solution degradation.

Q4: It is beneficial to include a comparison with other stabilizers reported in the literature, particularly in the discussion of improved film quality (e.g., grain size, crystallinity), to highlight the uniqueness of TFPH.

Response: We sincerely appreciate the reviewer for raising this important point. We have compared our work with other reported stabilizers and summarized the results in **Table R1** (**Table S2** in Supporting Information) on multi-parameters on film quality, including grain size, XRD full width at half maximum (FWHM), and champion PCE for five state-of-the-art solution stabilizers in comparison with TFPH. The data show that TFPH-modified films deliver the larger grains, and the narrowest XRD peaks, translating into the highest PCE among the compared single-additive systems, thereby underscoring its distinct effectiveness. We have now added the related discussion in the revised manuscript:

“Compared to the reported stabilizers that enhance the quality of perovskite films, including increasing grain size and reducing FWHM of XRD diffraction peaks, our results

rank among the best performers (**Table S2**), further demonstrating the effectiveness of the TFPH additive.” (Page 10, lines 6-9)

Table R1 (Table S2). Comparison of TFPH and other additives in film characteristics and PCE.

Additive	Grain size / nm	XRD FWHM /°	PCE / %	References
1-(3-(trifluoromethyl)phenyl) thiourea	319.4	0.233	24.7	Adv. Funct. Mater. 2025 , 35(5): 2414423.
saccharin sodium	NA	0.090	24.8	Small. 2025 , 2503279.
trifluorophenylacetic acid	1250	0.95	24.56	InfoMat. 2023 , 5(10): e12423
phenylhydrazine-4-sulfonic acid	3000	0.07	25.10	ACS Energy Lett. 2024 , 9(6): 2790-2799.
sulfaguanidine	124	NA	24.34	Adv. Funct. Mater. 2024 , 34(46): 2407897.
4-(trifluoromethyl) phenylhydrazine	590	0.084	26.0	This work

Q5: It is better to include a comparison of films and devices properties (TOF-SIMS, XRD, etc) before and after long-term sunlight stability tests to highlight and provide more insights into their unique TFPH.

Response: We sincerely appreciate the reviewer for raising this important suggestion. Accordingly, we conducted additional experiments to compare the changes of several properties on the perovskite films before and after long-term sunlight stability tests for 200 hours.

Firstly, we used scanning electron microscopy (SEM) to analyze surface morphology changes of the above two samples with/without TFPH (**Figure R2**, **Figure S14** in Supporting Information). It is shown the perovskite film with TFPH (the target sample)

maintained a smooth and intact surface before and after long-term sunlight stability tests. However, the control film exhibited obvious pinholes and rough textures after light soaking.

Figure R2 (Figure S14). SEM shows the morphology of the control and target perovskite films before and after 200 hours of light soaking. The control (aged) sample shows distinguish difference on surface morphology comparing to the control (fresh) sample, suggesting fast degradation; while the target (aged) shows no difference comparing to the target (fresh) sample, indicating a quite stable feature.

Secondly, we conducted X-ray diffraction (XRD) measurements on these perovskite films and the results is presented in **Figure R3 (Figure S15** in Supporting Information). There is no evident change on the XRD peak intensity or new peaks for the target film after 200 h light soaking, indicating the ultra-stable perovskite crystal structure. In contrast, the control sample exhibited an intense PbI_2 peaks after light soaking.

Figure R3 (Figure S15). XRD shows the structure of the control and target perovskite films before and after 200 hours of light soaking. Apparent PbI_2 peak appears in the control (aged) sample.

Thirdly, we carried out UV-Vis spectra on the above two samples in **Figure R4a (Figure S16a in Supporting Information)**. It is shown that there is no obvious change on the light absorption for the target sample before and after long-term sunlight stability tests. However, there is a significant decrease on light absorption for the control sample before and after long-term sunlight stability tests.

Finally, we used steady-state photoluminescence (PL) to evaluate the quality of these perovskite films. As illustrated in **Figure R4b (Figure S16b in Supporting Information)**, the control film exhibits a significantly reduced PL emission intensity after long-term sunlight stability test, suggesting high defect generation after 200 h light soaking. Encouragingly, the PL intensity of the target film only shows a slight decrease. This result suggests that TFPH can well preserve the structure of perovskite film during light soaking, preventing phase separation and defect generation in the perovskite film.

Figure R4 (Figure S16). Optical properties of the control and target perovskite films before and after 200 hours of light soaking. a) UV-Vis spectra. b) Steady-state PL spectra.

Therefore, these characterization results collectively demonstrate that TFPH can effectively inhibit the degradation of perovskite films under prolonged illumination, providing direct evidence for its role in improving the stability of PSC devices.

We have added the related discussion into the revised manuscript:

“To gain an in-depth understanding of the additive’s stabilizing effect, we systematically characterized the control and target perovskite films before and after 200 h of light soaking. These comprehensive analyses (Figure S14-S16) consistently confirm that TFPH effectively stabilizes the target perovskite films, in line with the observed excellent operational stability.” (Page 14, line 20-Page 15, line 1)

Response to Reviewer #2:

The authors report a stabilization strategy of perovskite precursor by incorporating a multifunctional molecule of TFPH. They first reveal the precursor degradation process in air, and then show the beneficial roles of TFPH in inhibiting solution degradation and improving the film quality of perovskite films. They finally demonstrated devices with efficiencies of around 26% and decent stability by utilizing solutions after storage in the glovebox for two months.

The proposed strategy of using TFPH additive is effective in inhibiting the precursor degradation, however, the reviewer found no new or advanced understanding of the

precursor degradation mechanism compared with that demonstrated in previous reports (Science Advance, 2021 7, eabe8130). In addition, the mechanism of TFPH-inhibited precursor degradation seems also no obvious difference with that reported in previous work (Science Advance, 2021 7, eabe8130). As two key findings in the present work show no obvious advancements, the reviewer considers the manuscript not suitable for publication in Nature Communications. Some detailed comments are listed below:

Response: We sincerely appreciate the reviewer's positive evaluation of our work in addressing the dual-phases stability of perovskite. The recognition of achieving a PCE of ~26% with decent stability (2-month glovebox storage) encourages us to in-depth explore the fundamental mechanisms.

In response to the reviewer's comments, we have conducted a comprehensive re-examination and analysis of our experimental data and results. Below, we will highlight the key distinctions and advancements of our work compared to the aforementioned SA study, with particular emphasis on novel mechanistic insights into perovskite stabilization.

Table R2. The list of performance parameters of the PSC samples from SA work and our work.

PSC devices	Modified samples	SA work	Our work	Increasing
Unmodified	Champion PCE	21.8%	23.71%	8.7%
	Batch uniformity (fresh)	17.6-21.8%	22.99±0.40 %	
	Batch uniformity (aged)-2 months	13.6-19.2 %	16.82±2.07 %	
	Batch consistency	NA	±0.40 - 2.07 %	
	Operational Stability (aged)	Encapsulated, T ₈₀ = ~500 h	Encapsulated, T₈₀ > 1200 h	
Modified with BHC/TFPH	Champion PCE	23.4%	26.0%	11%
	Batch uniformity (fresh)	NA	25.66±0.14%	
	Batch uniformity (aged)-2 months	21.4-23.3%	25.69±0.15%	

	Batch consistency	NA	$\pm 0.14 - 0.16\%$	
	Operational Stability(aged)	Encapsulated, No degradation for 1000 h while big fluctuation.	Encapsulated, T_{92} = ~1830 h with gradual decrease on PCE.	

We have systematically compared our results with prior art (the mentioned SA work) in the revised manuscript (**Table R2**). Our comprehensive data demonstrate: (1) ~11% relative PCE improvement for the modified PSC devices (2) ~1.8 x operational stability without big fluctuation (**Figure R5**), and (3) dramatically improved batch uniformity and consistency (standard variation on PCE < 0.17%). Comparably, the mentioned SA work demonstrated a pioneer while primary modified effect on the performance of PCE devices and uniformity, leaving the issue of batch consistency unstudied.

Figure R5. Long-term operational stability under maximum power point tracking (MPPT) in a) SA work. b) this work. There is evident fluctuation on PCE for the SA work devices while much smaller in our work.

Moreover, while our modification strategy represents a fundamental advancement over previous works in terms of effectiveness (achieving higher efficiency, superior stability, and addressing the core issues more comprehensively), we acknowledge the reviewer’s valid point that we did not sufficiently highlight the novel mechanisms underlying our stabilization approach. We sincerely appreciate the reviewer for identifying this shortcoming in our presentation, as it provides us with a valuable opportunity to clarify the innovative progress and scientific breakthroughs of our work.

1. Process Reliability & Reproducibility: Our study employs a well-established spin-coating method, which provides robust control over device performance and batch-to-batch consistency (uniformity and reproducibility). This approach allows us to clearly attribute performance enhancements to the introduced stabilizer. In contrast, the SA work utilized blade coating—a technique that, as of year 2021 and even presently, remains less mature for perovskite devices. This methodological difference may introduce significant variability in device uniformity, potentially obscuring the true effectiveness of their stabilization strategy.

2. Key Differences in Stabilizer Introduction: Critically, our stabilizer incorporation strategy fundamentally differs from that of the SA study. We directly introduce the stabilizer into the fresh perovskite precursor solution, enabling in situ stabilization prior to aging. This approach (1) promptly mitigates I_2 -induced degradation and (2) stabilizes the precursor solution from the outset. In contrast, the SA work adds stabilizers post-aging, which may limit their effectiveness. Moreover, as we reveal in our proposed mechanism (given below in the response to Q1), our stabilizer (TFPH) not only scavenges more I_2 but also releases N_2 —a byproduct that further shields the solution from ambient moisture/oxygen. This also provides us with new inspiration: the degradation of perovskite precursors can be suppressed by bubbling nitrogen or even hydrogen, a strategy we will explore in our follow-up research.

3. Novel Mechanistic Insights (TFPH Stabilizer): Most importantly, we elucidate the mechanism underlying TFPH’s exceptional performance: According to our comprehensive characterization and DFT calculations (**Figure R7** and **Figure R9**), a dual-pathway I_2

scavenging mechanism for TFPH to stable dual-phased perovskite is proposed and can be attributed as the strategic integration of hydrazine (-NH-NH₂) and trifluoromethyl (-CF₃) groups (**More details, please refer to the response to Q1**): The hydrazine group reduces I₂ to I⁻ at low concentrations, while directly incorporating I₂ into TFBI (with N₂ as byproduct, **Figure S4**) at high concentrations (**Figure 2e**, reactions (4) and (5)). Moreover, -CF₃ improves thin-film quality through dipole modulated crystallization, hydrophobicity, and defect passivation. These advantages are beneficial for the initial performance of the fabricated PSCs and more importantly ensure remarkable consistency in device performance across multiple production batches as being demonstrated in our work.

We believe the above innovative progress and mechanistic insights provide substantial contributions to the field of perovskite photovoltaics, advancing the understanding on multifunctional additive design principle for stable perovskite materials and boosting the batch stability and consistency based on high-performance PSCs.

Q1: The authors characterized molecular interactions between the TFPH molecule and I₂ to understand the mechanism of the inhibited precursor degradation. It is clear that the TFPH possesses two key functional moieties of hydrazine and trifluoromethyl; the authors need to separately discuss the role of the two functional groups in stabilizing the precursor solution. In addition, previous work (*Science Advance* **2021** 7, eabe8130) using similar additives of benzyl/propyl/(2-thienylmethyl) hydrazine hydrochloride has confirmed the role of hydrazine groups in stabilizing the precursor solution, is there any new mechanism for the TFPH in the present work? These questions, which are important for clarifying the rationale of the work, must be properly addressed.

Response: We sincerely appreciate the review's insightful comments, which are crucial for clarifying the novelty and mechanistic depth of our work. As you noted, the previous SA study demonstrated that hydrazine groups (-NH-NH₂) can reduce reactive iodine species (e.g., I₂) to iodide (I⁻) via redox chemistry, thereby mitigating precursor solution degradation. However, our work reveals an unreported dual-pathway mechanism for I₂ suppression by the hydrazine group in TFPH (as shown in **Figure R6**):

Reduction Pathway at Low I₂ Concentrations: When I₂ levels are low (the I₂/TFPH ratio is less than 4:1), the hydrazine group (-NH-NH₂) in TFPH reduces I₂ to I⁻, consistent

with prior reports on hydrazine-based additives (Angew. Chem. Int. Ed. 2022, 61, e202206914). This mechanism was confirmed by NMR spectroscopy of TFPH/I₂ mixtures (**Figures 2a** and **2b**), with the reaction process illustrated in **Figures 2d** (4) and **2e** (4). Notably, we found that this reduction reaction occurs when I₂ concentrations are relatively low—a new observation not reported in the mentioned SA study (Sci. Adv. 2021, 7, eabe8130).

Figure R6. The reactions between I₂ and a) BHC. b) TFPH.

Nucleophilic Substitution Pathway at High I₂ Concentrations: Under excess I₂ conditions, TFPH directly scavenges I₂ through a nucleophilic substitution reaction, forming stable trifluoromethyl benzyl iodide (TFBI) and releasing inert N₂ gas (**Figures 2a, 2b, 2d** (5), and **2e** (5)). This pathway efficiently eliminates surplus I₂ without generating harmful byproducts, offering superior stabilization compared to conventional redox-based approaches. The released N₂ gas further protects the perovskite from moisture and oxygen degradation (**Figure S4**). This nucleophilic substitution pathway is also undisclosed in the mentioned SA study (Sci. Adv. 2021, 7, eabe8130).

Figure R7 (Figure 2). Underlying mechanism of inhibiting solution ageing by reducing I₂. a) ¹H NMR spectra of TFPH and b) ratio of ¹H NMR peak area at different I₂/TFPH ratios. c) Mass spectrum of I₂/TFPH in MeOH. d) Photographs of the reaction results between I₂ and TFPH. e) Proposed reaction process of TFPH to inhibit the perovskite solution ageing.

Although the trifluoromethyl (–CF₃) group does not directly scavenge I₂, it critically governs the formation of high-quality perovskite films through three key functions:

Improved Hydrophobicity: Contact angle measurements (**Figure R8, Figure S12** in Supporting Information) confirm that –CF₃ enhances moisture resistance, aligning with reports on fluorinated additives boosting device stability (NSR, DOI:10.1093/nsr/nwaf211).

Figure R8 (Figure S12). Contact angle of water on perovskite films: a) the control without TFPH b) the target with TFPH.

Defect Passivation: The electron-withdrawing nature of –CF₃ mitigates halide vacancies and suppresses non-radiative recombination (ACS Appl. Mater. Interfaces 2022, 14, 3930; Energy & Fuels 2023, 37, 667–674).

Crystallization Optimization: To better elucidate the influence of the –CF₃ on the overall molecular properties, we conducted DFT calculations to compare the dipole moments of phenyl hydrazine (PH) and TFPH. Compared to PH, TFPH has much larger dipole moment (**Figure R9**, Supporting Information), which promotes the oriented perovskite growth, favorable for the strain releasing (**Figure 3d**), as also observed in fluorinated ammonium salt-modified PSCs (*Nature* 2022, 603, 73). Besides, the decrease in contact angle between the perovskite precursor and SAMs indicates favorable wettability, which is attributed to the strong dipolar nature of TFPH (**Figure S10**, Supporting

Information). Enhanced wetting properties of perovskite precursor solutions on ITO substrates promote the fabrication of compact, defect-free perovskite thin films with high quality.

Figure R9 (Figure S10). The dipole moment of a) phenyl hydrazine (PH) and b) 4-(trifluoromethyl) phenyl hydrazine (TFPH). The dipole moment of TFPH is 2.5 times greater than that of PH.

Figure R10 (Figure S11). Contact angle of perovskite precursor solutions on SAMs modified ITO surface: a) the control (perovskite precursor without TFPH) b) the target (perovskite precursor with TFPH).

Figure R11 (Figure 3). Characterizations of the studied perovskite films. a) Cross-sectional SEM images. b) XRD patterns. c) Enlarged XRD areas (13.3 - 25°) with the calculated peak ratio of (001)/(111). d) Strain analysis based on the 2θ - $\sin^2\phi$ method. e) Dark I-V curves of the hole-only devices. f) Steady-state PL spectra. g) Time-resolved PL decays.

In summary, the above study demonstrates that the hydrazine group in TFPH suppresses I_2 through a dual-pathway mechanism, as rigorously validated by our comprehensive and in-depth experimental investigations. Simultaneously, $-\text{CF}_3$ improves thin-film quality through dipole modulation, hydrophobicity, and defect passivation. The integration and synergistic interplay of these two functional groups endow TFPH with the above multifunctionality. This work establishes a novel paradigm for molecular additive design toward highly efficient and stable perovskite photovoltaics. Your thoughtful questions have helped us better articulate the scientific rationale of this work, and we are deeply grateful for your expertise.

Accordingly, we add the separated discussion on the role of the two functional groups ($-\text{NH}-\text{NH}_2$ and $-\text{CF}_3$) in stabilizing the precursor solution and boosting the performance of batch stability and consistency.

“Based on the comprehensive characterization above, the mechanism by which TFPH stabilizes dual-phase perovskite and enhances the quality of perovskite films can be attributed to the synergistic integration of the hydrazine ($-\text{NH}-\text{NH}_2$) and trifluoromethyl ($-\text{CF}_3$) functional groups: The hydrazine group reduces I_2 to I^- at low concentrations while directly incorporating I_2 into TFBI at high concentrations (**Figure 2**). The released byproduct, N_2 gas, further shields the perovskite from moisture and oxygen-induced degradation (**Figure S4**). Meanwhile, the $-\text{CF}_3$ group improves thin-film quality through multiple mechanisms: (1) dipole-modulated crystallization, which promotes preferential orientation and facilitates strain relaxation (**Figure S10**); (2) enhanced wetting of perovskite precursors on SAMs-modified ITO surfaces (**Figure S11**); and (3) effective defect passivation coupled with increased hydrophobicity (**Figure S12**). The synergistic effects of these two functional groups in TFPH not only benefit the initial performance on both efficiency and stability of the fabricated PSCs but also ensure exceptional batch-to-batch reproducibility in device performance, as demonstrated below.” (Page 11, line 29-Page 12, line 14)

Q2: In Figure 1 d and e, the ‘Intensity (a.u.)’ for the y-axis of the NMR results seems unnecessary.

Response: We sincerely appreciate the reviewer's professional suggestion. Following this advice, we have updated **Figures 1d** and **1e** (as shown in **Figure R12**). In the revised figures for the NMR results, the y-axis label “Intensity (a.u.)” has been removed.

Figure R12 (Figure 1). Investigation of the ageing of FAI and perovskite solutions. a) ^1H NMR spectra and b) the corresponding UV-Vis spectra of the fresh and aged FAI solutions in air. DMSO- d_6 is used for all ^1H NMR tests. The inset is the photograph of the FAI solutions in air. c) UV-Vis spectra of perovskite solutions. ^1H NMR spectra of the d) control and e) target perovskite solutions exposed to air for 1-3 days. (f) The percentage reduction of the integral area of c_2 and H_2O were obtained from the ^1H NMR spectra.

Q3: The authors presented device results utilizing precursors with different aging time in the glovebox, it would be necessary to present the performance evolution for the two types of precursors with different aging time in ambient air?

Response: We sincerely thank the reviewer for this insightful suggestion. To further evaluate the stability of our strategy in harsher conditions, we systematically investigated the performance evolution of devices fabricated from precursor solutions aged in ambient air (26.7 $^{\circ}\text{C}$, 66% relative humidity). As shown in **Figure R13** and **Table R3-R4 (Figure S13 and Tables S8-S9** in Supporting Information), TFPH-based target devices exhibit superior stability compared to the control devices. When prepared from fresh solutions, the target devices achieve a champion PCE of 25.83%, outperforming the control devices (23.32%). Notably, after 1-day aging in air, the target devices retain a high PCE of 25.54% (only 0.2% loss), while the control devices degrade more significantly to 22.33% (0.8%

loss). This stability advantage persists under prolonged aging: after 2, 4, and 8 days, the target devices maintain PCEs of 24.91%, 23.88%, and 22.51%, respectively, whereas the control devices degrade rapidly to 20.71%, 18.32%, and 15.77%. These results robustly demonstrate that TFPH incorporation effectively mitigates precursor degradation, even under humid air conditions.

Importantly, this comparison also reveals that ambient moisture accelerates precursor degradation more severely than inert glovebox conditions, underscoring the critical challenge of stabilizing perovskite precursors in high-humidity environments. While our current strategy already shows promising moisture resistance, further studies to enhance the air stability of precursor solutions—particularly for scalable fabrication—remain an essential direction for future research.

These results and the corresponding discussion have been incorporated into the revised manuscript:

“We further compared device performance under harsher conditions, specifically ageing in ambient air (**Figure S13a**). Devices were fabricated from perovskite precursors with and without TFPH, using both fresh and air-aged solutions over several days. It is found that the stability advantage persists under prolonged aging: after 2, 4, and 8 days, the target devices maintain PCEs of 24.91%, 23.88%, and 22.51%, respectively, whereas the control devices degrade rapidly to 20.71%, 18.32%, and 15.77%. These results robustly demonstrate that TFPH incorporation effectively mitigates precursor degradation, even under humid air conditions (**Figure S13b** and **Table S8 and S9**).” (Page 14, lines 3-11)

Figure R13 (Figure S13). a) The precursor solution with and without TFPH doping aged in air, the temperature and relative humidity. b) Statistic data of perovskite solution, stored in air, at various ageing times.

Table R3 (Table S8). Photovoltaic parameters of the champion PSCs prepared with the control solution at different ageing times in air.

Time	PCE %	FF%	V_{oc} (V)	J_{sc} (mA cm ⁻²)
0	23.32	81.90	1.131	25.19
1	22.33	80.17	1.097	25.39
2	20.71	77.64	1.069	24.94
4	18.32	71.64	1.063	24.05

8	15.77	62.66	1.053	23.89
---	-------	-------	-------	-------

Table R4 (Table S9). Photovoltaic parameters of the champion PSCs prepared with the target solution at different ageing times in air.

Time	PCE %	FF%	V_{oc} (V)	J_{sc} (mA cm ⁻²)
0	25.83	85.42	1.167	25.91
1	25.54	85.68	1.163	25.65
2	24.91	83.37	1.156	25.84
4	23.88	82.71	1.153	25.05
8	22.51	80.90	1.118	24.89

Response to Reviewer #3:

This manuscript presents a highly significant and timely study on the stabilization of perovskite solar cells (PSCs) using 4-(trifluoromethyl)phenyl hydrazine (TFPH) as a solution-phase and solid-phase stabilizer. The work addresses a critical challenge in the field of perovskite photovoltaics—namely, the instability of perovskite precursor solutions and films—which is a major barrier to the commercialization of PSCs. The authors demonstrate a novel and effective approach to enhance both the storage stability of perovskite solutions and the operational stability of devices, achieving remarkable power conversion efficiencies (PCEs) of ~26.0% with excellent batch-to-batch consistency. The study is well-executed, with a strong emphasis on understanding the underlying mechanisms of TFPH's stabilizing effects, making it a valuable contribution to the field. I recommend this work for publication after addressing the following issues:

Response: We thank the reviewer for the positive assessment and for recognizing the importance of our work.

Q1: The abstract provides a good overview of the study, but it could benefit from a more detailed explanation of the specific mechanisms by which TFPH stabilizes the perovskite solution and enhances device performance. For instance, the abstract briefly mentions that TFPH inhibits degradation, promotes oriented crystallization, and reduces trap density, but it does not elaborate on how these effects are achieved. Including a few more sentences to explain these mechanisms would make the abstract more informative and compelling. Additionally, the abstract could highlight the broader implications of this work for the commercialization of PSCs.

Response: Thanks for your kind suggestion. To help readers directly and clearly grasp the key findings of this study, we have briefly elaborated the mechanisms by which stabilizes the perovskite solution and enhances device performance in the Abstract:

“These results are attributed to TFPH's multifunctionality: a) hydrazine groups inhibit perovskite decomposition; b) trifluoromethyl boosts dipole moment, aiding crystallization; c) impurity reduction and high-quality film jointly lower charge traps.” (Page 2, lines 11-15)

Q2: The introduction is well-structured and provides a solid background on the challenges of perovskite stability. However, it could benefit from a more detailed discussion of the current state of perovskite solar cell commercialization and how this study addresses specific industrial challenges. For example, the introduction could include a brief overview of the current limitations in large-scale production and how TFPH offers a solution to these problems. This would help readers better understand the significance of the work in the context of real-world applications.

Response: We thank the reviewer for the insightful suggestion. In response to the reviewer's suggestion, we have expanded Introduction to include a more detailed discussion of the current challenges of PSC commercialization and clarified how our study addresses the specific challenges relevant to industrial implementation as the reviewer mentions above.

“...The solution-based fabrication method offers significant advantages, including low-cost production, compatibility with large-area and flexible substrates, and scalability for industrial manufacturing. However, the instability of perovskite precursor solutions remains a critical challenge, severely limiting the reproducibility and consistency of device performance during large-scale production (*Nature* **2024**, 628, 299; *Science* **2024**, 386, 531).

Batch-to-batch variability induced by this instability poses a significant challenge to the commercialization of PSCs (*ACS Materials Letters* **2021**, 3, 351). As organic-inorganic hybrid materials, perovskites inherently exhibit relatively poor stability, with degradation occurring under both storage and operational conditions. The degradation of perovskite precursor solutions over time, influenced by many factors, including moisture, oxygen, and temperature, can result in substantial differences in the film formation, thereby introducing inconsistencies in the scale-up fabrication of PSCs (*Nature Communications* **2024**, 15, 4552). These issues underscore the urgent need for strategies to enhance the stability of both perovskite solutions and solid-state films.” (Page 3, lines 6-19)

“...To mitigate this issue, we introduced 4-(trifluoromethyl) phenyl hydrazine (TFPH) as a stabilizer to modify the perovskite solution, which significantly enhanced the stability of perovskite in both solution and solid phases. TFPH acts through multiple complementary mechanisms: stabilizing the perovskite lattice via redox-active hydrazine moieties, guiding preferential crystal orientation through dipolar interactions, and minimizing trap density by simultaneously removing impurities and high-quality film.” (Page 4, line 27-Page 5, line 3)

Q3: The manuscript lacks detailed experimental procedures, particularly in the preparation of perovskite solutions and device fabrication. Adding a more comprehensive methods section would improve reproducibility and allow other researchers to replicate the study. For example, the authors should provide specific details on the concentrations of TFPH used, the exact aging conditions, and the steps involved in device fabrication. This would enhance the transparency and reliability of the study.

Response: We sincerely thank the reviewer for pointing out this issue. We had prepared a comprehensive methods section in supporting information file, including Materials and methods, Preparation of TFPH₂PbCl₃ crystals, Perovskite precursor solution, Device fabrication, Device characterization, and Other characteristics. According to the reviewer’s suggestions, we add more specific details in the experimental part:

For the concentrations of TFPH used, we add “For the target precursor solution, 2.937 g FAI, 234 mg CsI, 8.381 g PbI₂ and 5 mg of TFPH were dissolved in 10 mL mixed solvents of DMF and DMSO (v/v, 4/1) with continuous overnight stirring.” (SI, Page 2, lines 26-28)

For the ageing conditions, we add: “The perovskite precursors for device fabrication were aged in an N₂-filled glovebox with oxygen levels <10 ppm and water levels <0.1 ppm.” (SI, Page 2, line 29- Page 3, line 1)

For the steps involved in device fabrication, it can refer to the “Device fabrication” part in SI, which has provided the whole procedure in details. “The planar p-i-n perovskite solar cells were fabricated with an architecture of ITO/ 2PADCBCS_{0.05}FA_{0.95}PbI₃/ C₆₀/BCP/Cu. Patterned ITO glass was ultrasonically cleaned for 20 min with a detergent, deionized water, acetone and ethanol, sequentially. Then, the ITO-coated glass substrates were treated with ultraviolet ozone (UVO) for 20 min after being dried by an N₂ gun, then the treated ITO glass was transferred to the glove box. 2PADCBC solution (0.5 mg mL⁻¹) in MeOH was spun onto the above ITO substrate at 3,000 rpm for 5 s, and then annealed at 100 °C for 5 min. The perovskite precursor solution was spread on the 2PADCBC coated substrate at 2,000 rpm for 10 s and then at 5,000 rpm for 30 s. Then, 130 μL of CB was quickly dropped in 10 s before the end of the procedure. The wet films were immediately transferred to a heating plate and annealed at 100°C for 30 min. A saturated PDI solution was spin-coated on the perovskite surface at 5000 rpm and annealed 100°C for 5 min as post-treatment. Devices were completed by the thermal evaporation of 40 nm C₆₀, 8 nm BCP, and 100 nm Cu.” (SI, Page 3, lines 3-16)

Moreover, we also provide detailed AFM characterization conditions in the experimental part: “The AFM images of surface morphology were conducted by Bruker Multimode 8 in air (RH 40%-50%, RT 25°C).” (SI, Page 4, lines 11-12)

For the dipole moment calculation, we add “The dipole moment of phenyl hydrazine and 4-(trifluoromethyl) phenyl hydrazine were calculated based on the density functional theory using the Gaussian 09 software package. The structure optimization was carried out at B3LYP/6-311g, and the calculated results were analyzed by the GaussView.” (SI, Page 4, lines 12-15)

We hope the above revision can improve reproducibility and allow other researchers to replicate our study.

Q4: While the PCE is highlighted, other important metrics such as fill factor (FF), open-circuit voltage (V_{oc}), and short-circuit current (J_{sc}) should be discussed in more details. For

instance, the authors could provide a more comprehensive analysis of how TFPH affects these parameters and how they contribute to the overall device performance. This would provide a more complete picture of the device's performance.

Response: Thanks for your advice. It is quite necessary to provide a comprehensive analysis of how TFPH affects these parameters and how they contribute to the overall device performance. This undoubtedly helps the reader to understand the completed picture of the device's performance.

All the photovoltaic parameters had been summarized in **Table R5-R6** (**Table S5** and **S6** in Supporting Information) and their trend with the aging time of precursor solution is provided in **Figure 4d**.

The photovoltaic parameters for the fresh champion devices were specifically given in **Figure 4b & 4c**. Compared to the control device, the target device has slight change on J_{sc} while significant increase on V_{oc} (from 1.121 to 1.186 V) and FF (from 0.826 to 0.857). The superior V_{oc} results from the improved crystallization and reduced trap density; the elevated FF from the compact and highly oriented perovskite grains and smooth surface, favoring the carrier transportation.

For the aged devices, the observed performance degradation is primarily attributed to the significant reduction in J_{sc} with ageing time. Specifically, the control devices exhibited a substantial decrease in J_{sc} from 25.38 to 20.68 mA cm⁻², accompanied by an increase in standard deviation from 0.42 to 1.16. In marked contrast, the target devices demonstrated excellent stability, maintaining J_{sc} values above 25.5 mA cm⁻² throughout the ageing process with minimal variation, indicating superior reproducibility. This performance stability can be ascribed to both the well-preserved precursor composition stabilized by TFPH.

We have added the above discussion in the revised manuscript as follow:

1. “The superior V_{oc} results from the improved crystallization and reduced trap density; the elevated FF from the compact and highly oriented perovskite grains and smooth surface, favoring the carrier transportation. Therefore, this above result is consistent with the better quality of the target perovskite film that is modified by TFPH (**Figure 3**).” (Page 12, lines 24-28)

2. “The observed performance degradation is primarily attributed to the significant reduction in J_{sc} with ageing time. Specifically, the control devices exhibited a substantial decrease in J_{sc} from 25.38 to 20.68 mA cm⁻², accompanied by an increase in standard deviation from 0.42 to 1.16. In marked contrast, the target devices demonstrated excellent stability, maintaining J_{sc} values above 25.5 mA cm⁻² throughout the ageing process with minimal variation, indicating superior reproducibility. This performance stability can be ascribed to the well-preserved precursor composition stabilized by TFPH.” (Page 13, lines 6-13).

Table R5 (Table S5). Photovoltaic parameters of PSCs prepared from the control solution at different ageing times.

Time	Data	PCE %	FF%	V_{oc} (V)	J_{sc} (mA cm ⁻²)
1	Average	22.99±0.40	81.55±0.92	1.114±0.00 6	25.38±0.42
	Champion	23.71	82.59	1.121	25.45
15	Average	21.39±0.74	79.39±0.95	1.117±0.00 3	24.37±0.51
	Champion	22.70	80.28	1.122	25.24
30	Average	19.58±1.12	77.67±2.24	1.124±0.00 6	23.01±0.93
	Champion	21.74	81.77	1.136	23.25
60	Average	16.82±2.07	76.06±2.98	1.118±0.00 9	20.68±1.61
	Champion	20.21	79.85	1.135	22.23

Table R6 (Table S6). Photovoltaic parameters of PSCs prepared from the target solution at different ageing times.

Time	Data	PCE %	FF%	V_{oc} (V)	J_{sc} (mA cm ⁻²)
1	Average	25.66 ± 0.29	84.77 ± 1.01	1.179 ± 0.008	25.66 ± 0.14
	Champion	25.95	85.70	1.186	25.53
15	Average	25.67 ± 0.31	84.69 ± 1.03	1.180 ± 0.006	25.68 ± 0.16
	Champion	25.98	85.72	1.186	25.56
30	Average	25.67 ± 0.33	84.92 ± 0.49	1.179 ± 0.007	25.65 ± 0.15
	Champion	26.00	85.35	1.185	25.71
60	Average	25.62 ± 0.29	84.69 ± 0.63	1.178 ± 0.006	25.69 ± 0.15
	Champion	25.91	85.32	1.184	25.64

Q5: The XRD patterns are mentioned, but a more detailed discussion of the crystallographic changes induced by TFPH would be beneficial. For example, the authors could provide a more in-depth analysis of the XRD data, including the crystal structure and orientation of the perovskite films. This would help readers understand how TFPH affects the crystallization process.

Response: We thank the reviewer for this constructive suggestion. To deepen the understanding of TFPH's impact on crystallization, we conducted a detailed analysis of XRD data, focusing on crystallographic orientation and crystallinity in revised manuscript: "The X-ray diffraction (XRD) patterns in **Figure 3b** show that all samples exhibit a prominent diffraction peak at 13.9°, corresponding to the (001) crystallographic plane of the perovskite, which indicates preferred in-plane growth parallel to the substrate. A comparative analysis of the intensity ratio between the (001) and (111) peaks (**Figure 3c**)

reveals that the TFPH-treated (aged) film has a higher $I_{(001)}/I_{(111)}$ ratio than both the control (fresh) and control (aged) films. This demonstrates that TFPH preserves and enhances the preferential (001) orientation even after precursor ageing. Additionally, the full width at half maximum (FWHM) of the (001) peak decreases from 0.092° (control, fresh) to 0.087° (control, aged) and 0.084° (TFPH-treated, aged), indicating progressively improved crystallinity in the TFPH-treated film. These results suggest that TFPH promotes more ordered nucleation and growth, contributing to compact and highly oriented perovskite grains. Compared to the reported stabilizers that enhance the quality of perovskite films, including increasing grain size and reducing FWHM of XRD diffraction peaks, our results rank among the best performers (**Table S2**), further demonstrating the effectiveness of the TFPH additive.” (Page 9, line 16 – Page 10, line 9).

Q6: A cost analysis of using TFPH in large-scale production would provide valuable insights into its commercial viability. For example, the authors could estimate the cost of TFPH synthesis and its impact on the overall cost of PSC production. This would help readers assess the economic feasibility of using TFPH in commercial devices.

Response: We thank the reviewers for their valuable suggestions. Below, we provide the cost estimation of perovskite solar cells and compare the economic benefits of the TFPH additive.

Table R7 (Table S11). List of raw material unit prices.

Materials	Price (USD)	Per unit
ITO glass	0.5	piece ($1.5 \times 1.5 \text{ cm}^2$)
FAI	15	g
PbI ₂	2	g
CsI	4	g
C ₆₀	167	g
BCP	470	g

Cu	1	g
TFPH	1	g
DMF	84	L
DMSO	575	L
CB	200	L

Based on the raw material quantities used in the fabrication process and the unit prices listed in **Table R7 (Table S11)** in Supporting Information), we estimate that the production cost of a control and target device to be \$0.784168 and \$0.784368, respectively.

Table R8 (Table S12). The cost-benefit relationship enabled by the TFPH additive.

	Control	Target	Comparison
Cost per 1 mL precursor (USD)	6.3573	6.3623	+0.079 %
Cost per device (USD)	0.784168	0.784368	+0.026 %
Average PCE (%)	22.99	25.66	+11.62 %

To better evaluate the cost-benefit relationship enabled by TFPH, we present the detailed comparison in **Table R8 (Table S12)** in Supporting Information). The incorporation of TFPH results in a negligible increase in material cost, only 0.005 USD per mL of precursor and 0.0002 USD per device ($1.5 \times 1.5 \text{ cm}^2$). But encouragingly, it leads to a significant 11.62% improvement in PCEs from 22.99% to 25.66%. This comparison clearly demonstrates the industrial viability and economic advantage of our TFPH additive.

We have added the related discussion in the revised manuscript:

“To further evaluate the practical viability of TFPH, we performed a detailed cost-benefit analysis (Tables S11 and S12), which revealed that the modified perovskite precursor incurs only a minimal cost increase of 0.005 USD per mL while achieving a substantial 11.62% improvement in power conversion efficiency (PCE). This compelling cost-to-performance ratio underscores both the industrial feasibility and economic advantage of TFPH integration in PSC fabrication.” (Page 16, lines 1-6)

Q7: The conclusion is concise but could be expanded to summarize the key findings and their implications for the field of perovskite solar cells more comprehensively. For example, the authors could discuss how their findings could be applied to other types of perovskite devices or how TFPH could be used in combination with other stabilizers to further improve performance. This would provide a more forward-looking perspective on the study.

Response: We sincerely appreciate the reviewers' suggestion to further comprehensively summarize the key findings and applications of this work, which will help highlight its potential value and guide future research directions. This valuable input will encourage more researchers in the field to join the efforts in enhancing the stability and consistency of perovskite device batches.

To verify the universal effectiveness of the TFPH-based strategy in enhancing device performance, we meticulously incorporated TFPH into the precursors of both narrow-bandgap and wide-bandgap perovskites. For wide-bandgap perovskite (1.75 eV), we fabricated devices with an identical structure of FTO/SAMs/perovskite/C₆₀/BCP/Cu. As depicted in **Figure R14a** (**Figure S17a** in Supporting Information), the control device impressively achieved a PCE of 20.3%, characterized by a V_{OC} of 1.306 V, a J_{SC} of 18.6 mA cm⁻², and an FF of 83.5%. In contrast, upon the strategic introduction of TFPH into the precursor solution, the target device demonstrated a champion PCE to 21.8%. This improvement was accompanied by a V_{OC} of 1.342 V, a J_{SC} of 18.8 mA cm⁻², and an FF of 86.2%, clearly indicating the positive impact of TFPH. Similarly, for narrow-bandgap perovskite (1.25 eV), we prepared devices using the structure of FTO/PEDOT:PSS/perovskite/C₆₀/BCP/Cu. As illustrated in **Figure R14b** (**Figure S17b** in Supporting Information), the control device reached a champion PCE of 21.3%, featuring a V_{OC} of 0.826 V, a J_{SC} of 33.3 mA cm⁻², and an FF of 77.5%. When TFPH was integrated, the target device

achieved a superior PCE of 22.6%, alongside a V_{OC} of 0.845 V, a J_{SC} of 33.4 mA cm^{-2} , and an FF of 80.1%. These experimental results firmly verify the broad effectiveness of TFPH in boosting the device performance across different types of perovskites, offering valuable insights for future research in this field.

We have added these discussions in the revised manuscript:

“Furthermore, we successfully incorporated TFPH into both narrow-bandgap (1.25 eV) and wide-bandgap (1.75 eV) perovskites, observing consistent performance enhancements in the corresponding devices, as confirmed by their improved current density-voltage (J-V) characteristics (**Figure S17**). These results highlight the universal efficacy of TFPH as a versatile additive, suggesting its broad applicability in optimizing perovskite solar cells (PSCs) with different bandgaps—an important step toward their commercialization.” (Page 15, line 13-Page 16, line 1)

Figure R14 (Figure S17). J-V curves of a) wide-bandgap perovskite solar cell. b) narrow-bandgap perovskite solar cell.

According to the reviewer’s suggestion, we rewrite the Conclusion part by integrating the part that how our strategy could be applied to other types of perovskite devices and how TFPH could be used in combination with other stabilizers to further improve performance. “This work reveals that oxidative degradation of FA-rich perovskite precursors in air—particularly I_2 generation—severely compromises film quality and device performance. By introducing TFPH as a multifunctional stabilizer, we simultaneously suppress solution-phase decomposition and elevate the quality of resulting perovskite films. The optimized devices achieve consistent PCEs of ~26.0% with exceptional operational stability, overcoming

batch-to-batch variability. The universal effectiveness of this approach is demonstrated through its successful application to normal, narrow- and wide-bandgap perovskite compositions, establishing TFPH's ability to mitigate moisture- and oxygen-induced degradation pathways. Beyond its standalone effectiveness, TFPH's molecular design suggests strong potential for synergistic combinations with other stabilizers—such as polymer additives or interfacial modifiers—to further enhance device performance and longevity. Our work provides both fundamental insights into perovskite stabilizing mechanisms and a practical, scalable strategy for industrial PSC production.” (Page 16, lines 8-21).

Response Letter to Reviewers

We are delighted that all the reviewers have endorsed our responses and recommended acceptance of our manuscript. We sincerely appreciate their recognition of our work's innovation and their valuable suggestions for final improvements prior to publication. Below we have carefully addressed all remaining points, and believe the revised manuscript is now in its final acceptable form. All revisions are highlighted in the updated manuscript.

Response to Reviewer #1:

The revised manuscript can now be accepted.

Response: We sincerely appreciate the reviewer's recommendation on our revised manuscript.

Response to Reviewer #2:

My key concern was with the new mechanism/understanding of stabilizing the precursor by using the presented TFPH and reported BHC featuring a similar functional moiety and reaction path. In the revised manuscript, the authors provided a detailed comparison and interpretation of the different aspects of TFPH, demonstrating the advantages of the TFPH and making the mechanistic understanding clear. As the BHC is functional similarly on stabilizing the precursor solution, it would be more appropriate to include the work in the third paragraph of the introduction. In addition, the key difference on the new mechanism compared with the BHC should also be included in the discussion part, which would be respectful to previous work. With these changes, the work would be considerable for publication.

Response: We sincerely appreciate the reviewer's recognition of our revised manuscript and their constructive suggestions for further improvement. We are grateful for the positive assessment of our detailed comparison between TFPH and BHC, as well as the clarification of the mechanistic understanding. We fully agree that placing this discussion in the proper

context will strengthen both the introduction and mechanistic discussion. Regarding the two specific suggestions:

Inclusion of BHC in the Introduction

We have incorporated the BHC reference (Science Advance, 2021 7, eabe8130, ref. 11) in the third paragraph of the introduction (now page 5, lines 15-17), “For example, hydrazine-based compounds, such as benzyl hydrazine hydrochloride (BHC), have been employed as additives to scavenge iodine species (I_3^-) from precursor solutions via redox reactions.¹¹”, and in the fourth paragraph of the introduction “To mitigate this issue, we developed 4-(trifluoromethyl) phenyl hydrazine (TFPH), a derivative of BHC,¹¹ as a multifunctional additive to modify the perovskite precursor solution” (now page 5, line 29-page 6, line 1), properly highlighting it as pioneering work in precursor stabilization. We have prominently cited Reference 11 twice in the Introduction section and referenced it more than three times throughout the manuscript.

Clarification of Key Mechanistic Differences in the Discussion

We have now expanded the Discussion section to explicitly outline the key distinctions between TFPH's stabilization mechanism and that of BHC as follow:

“It is demonstrated that hydrazine groups ($-NH-NH_2$) in BHC can reduce reactive iodine species (e.g., I_2) to iodide (I^-) via redox chemistry,¹¹ thereby mitigating precursor solution degradation. However, our work reveals an unreported dual-pathway mechanism for I_2 suppression by the hydrazine group in TFPH. 1) Reduction Pathway at Low I_2 Concentrations: When I_2 levels are low (the I_2 /TFPH ratio is less than 4:1), the hydrazine group ($-NH-NH_2$) in TFPH reduces I_2 to I^- , consistent with prior reports on hydrazine-based additives.^{9, 11} This mechanism was confirmed by NMR spectroscopy of TFPH/ I_2 mixtures (Figures 2a and 2b), with the reaction process illustrated in Figures 2d (4) and 2e (4). Notably, we found that this reduction reaction occurs when I_2 concentrations are relatively low—not reported in the work with BHC as stabilizer.¹¹ 2) Nucleophilic Substitution Pathway at High I_2 Concentrations: Under excess I_2 conditions, TFPH directly scavenges I_2 through a nucleophilic substitution reaction, forming stable trifluoromethyl benzyl iodide (TFBI) and releasing inert N_2 gas (Figures 2a, 2b, 2d (5), and 2e (5)). This pathway

efficiently eliminates surplus I_2 without generating harmful byproducts, offering superior stabilization compared to conventional redox-based approaches. The released N_2 gas further protects the perovskite from moisture and oxygen degradation (Figure S4). This nucleophilic substitution pathway has not been disclosed or characterized in earlier studies.^{9, 11}” (Page 9, lines 7-24)

We believe these modifications provide readers with a clearer evolutionary perspective while maintaining focus on our work's novel contributions. We believe this addresses the reviewer's concern about scholarly context while strengthening the manuscript's technical rigor. The tracked changes are highlighted in the revised manuscript for easy reference.

Response to Reviewer #3:

The authors have revised this manuscript following reviewers' comments, it can be accepted as the present form.

Response: We are glad to see that the reviewer #3 is satisfied with our revision and the acceptance recommendation of our work.